# Impact of urbanization on gas-phase pollutant concentrations: a regional scale, model based analysis of the contributing factors

Peter Huszar[1], Jan Karlický[1], Lukáš Bartík[1], Marina Liaskoni[1], Alvaro Patricio Prieto Perez[1], and Kateřina Šindelářová[1]

[1]Department of Atmospheric Physics, Faculty of Mathematics and Physics, Charles University, Prague, V Holešovičkách 2, 18000, Prague 8, Czech Republic

**Correspondence:** P. Huszar (peter.huszar@matfyz.cuni.cz)

**Abstract.** Urbanization or rural-urban transformation (RUT) represents one of the most important anthropogenic modifications of land-use. To account for the impact of such process on air-quality, multiple aspects of how this transformation impacts the air has to be accounted for. Here we present a regional scale numerical model (regional climate models RegCM and WRF coupled to chemistry transport model CAMx) based study for present day conditions (2015-2016) focusing on a range of central European cities and quantify the individual and combined impact of four potential contributors. Apart from the two most studied impacts, i.e. the urban emissions and the urban canopy meteorological forcing (UCMF, i.e. the impact of modified meteorological conditions) we focus also on two less studied contributors to the RUT impact on air-quality: the impact of modified dry-deposition due to transformed landuse and the impact of modified biogenic emissions due to urbanization induced vegetation modifications and changes in meteorological conditions affecting these emissions. To quantify each of these RUT contributors, we performed a cascade of simulations with CAMx driven with both RegCM and WRF were each effect was added one-by-one while we focused on gas-phase key pollutants: nitrogen and sulfur dioxide ($NO_2$ and $SO_2$) and ozone ($O_3$).

The validation of the results using surface observations showed an acceptable match between the modelled and observed annual cycles of monthly pollutant concentrations for $NO_2$ and $O_3$ while some discrepancies in the shape of the annual cycle were identified for some of the cities for $SO_2$ pointing to incorrect representation of the annual emission cycle in the emissions model used. The diurnal cycle of ozone was reasonably captured by the model.

We showed on an ensemble of 19 central European cities that the strongest contributors to the impact of RUT on urban air-quality are the urban emissions themselves, resulting in increases concentrations for nitrogen (by 5-7 ppbv on average) and sulfur dioxide (by about 0.5-1 ppbv), and, decreases for ozone (by about 2 ppbv), and the urban canopy meteorological forcing resulting in decreases of primary pollutants (by about 2 ppbv for $NO_2$ and 0.2 ppbv for $SO_2$) and increases of those of ozone (by about 2 ppbv). Our results showed that they have to be accounted for simultaneously as the impact of urban emissions without considering UCMF can lead to overestimation of the emission impact. Additionally, we quantified two weaker contributors: the effect of modified landuse on dry-deposition and the effect of modified biogenic emissions. Due to modified dry-deposition, summer (winter) $NO_2$ increases (decreases) by 0.05 (0.02) ppbv while almost no average effect for $SO_2$ in summer and a 0.04 ppbv decrease in winter is modelled. The impact on ozone is much stronger and reaches a 1.5 ppbv increase on average. Due

to modified biogenic emissions, negligible effect on $SO_2$ and winter $NO_2$ is modelled, while for summer $NO_2$, an increase by about 0.01 ppbv is calculated. For ozone, we found a much larger decreases between 0.5-1 ppbv.

In summary, when analyzing the overall impact of urbanization on air-pollution for ozone, the four contributors has the same order of magnitude and none of them should be neglected. For $NO_2$ and $SO_2$, the contribution of the landuse induced modifications of dry-deposition and the modified biogenic emissions have by at least one order smaller effect and the error will be thus small if they are neglected.

## 1 Introduction

Urbanization represents one of the most important transformations of land-use turning the natural surface into an built-up surface with objects like buildings, streets, roads etc. While urban areas represent only less than a percent of the total Earth surface (Gao et al., 2020), already more than half of the earth's population live in cities (UN, 2018a) and this transformation, which is often called rural-urban transformation (RUT) is still an ongoing process. It is expected that in the upcoming decades, more then 60% of the population will live in urban areas (UN, 2018b), making the research focusing their environmental effects more and more crucial.

It has been known that urban areas affect predominantly the atmospheric environment (Folberth et al., 2015) and they act via two primary intrusions that urbanization represents within the natural environment: i) it is the introduction of urban land-surface replacing rural one causing significant modifications of the meteorological conditions (Oke et al., 2017) and climate (Huszar et al., 2014; Zhao et al., 2017), and ii) the introduction of a massive emissions source of anthropogenic pollutants perturbing not only local but also regional and global air composition (Lawrence et al., 2007; Timothy and Lawrence, 2009; Im and Kanakidou, 2012; Huszar et al., 2016a) .

As for the air-quality of urban areas and those surrounding large cities, it is clear that the main driver affecting the concentrations are the local urban emissions. Indeed, many studies looked on the perturbation of the atmospheric composition due to solely the urban emissions over different scales. For example, Lawrence et al. (2007), Butler and Lawrence (2009) or Stock et al. (2013) investigated the global impact of emissions from megacities, while on regional scales many studies focused on large agglomerations in Europe, like Athens, Istanbul, London or Paris (e.g. Im et al., 2011a, b; Im and Kanakidou, 2012; Finardi et al., 2014; Skyllakou et al., 2014; Markakis et al., 2015; Hodneborg et al., 2011; Huszar et al., 2016a; Hood et al., 2018) or on large eastern Asian pollution hot-spots (Guttikunda et al., 2003, 2005; Tie et al., 2013). These studies shown that, not surprisingly, the concentrations of primary pollutants like oxides of nitrogen and sulfur (NOx, $SO_2$), volatile organic compounds, primary aerosols are substantially increased. But on the other hand, urban emissions, due to their high NOx-to-VOC ratio can lead to decreases of ozone in the urban cores (e.g. Huszar et al., 2016a). There is further a general consensus in these studies, that although air pollution in cities is determined mainly by the local sources, significant fraction of the total concentration is associated to rural sources or to sources from other cities (Panagi et al., 2020; Thunis et al., 2021; Huszar et al., 2021).

Urbanization however influences the final air pollution other ways too. One of the most studied aspect of RUT is the modulation of the pollutant concentration due to the meteorological forcing represented by the urban canopy which includes effects

like increased temperatures (urban heat island, UHI) (Oke, 1982; Oke et al., 2017; Karlický et al., 2018; Karlický et al., 2020; Sokhi et al., 2022), lower wind-speeds (Jacobson et al., 2015; Zha et al., 2019) or elevated boundary layer height along with enhanced vertical eddy diffusion Ren et al. (2019); Huszar et al. (2020a); Wang et al. (2021b). Huszar et al. (2020a) introduced the term urban canopy meteorological forcing (UCMF) which represents the forcing that the land-surface modified by the RUT represents on the physical state of the air above via perturbed exchange of momentum, heat, radiation and moisture. UCMF thus is a modification of meteorological conditions which in turn propagates to modifications on pollutant concentrations via modifying the transport, chemical transformation and deposition of air pollutants. Indeed, Ulpiani (2021) argued that the urban pollution has to be studied in connection with the UHI and other related meteorological effects. Many other studies looked at impact of UCMF on air-quality and found that the most important parameters in this regard are temperature, turbulence and wind (Struzewska and Kaminski, 2012; Liao et al., 2014; Kim et al., 2015; Zhu et al., 2017; Zhong et al., 2018; Li et al., 2019; Huszar et al., 2014, 2018a, 2020b) while moisture effects were rather minor (Huszar et al., 2018b). These studies found, that these changes led to near-surface decrease of primary pollutant concentrations while in case of secondary pollutants (e.g. ozone) increases are encountered either on the surface or at higher levels (Huszar et al., 2018a; Janssen et al., 2017; Yim et al., 2019; Li et al., 2019; Kim et al., 2021; Kang et al., 2022). In other words, besides the urban emission input, UCMF is another factor that contributes to the final urban pollution within the overall process of RUT (Huszar et al., 2021).

Moreover, during urbanization the land-use is modified from rural (or natural like forest/grassland etc.) to "urban" one which itself introduces a forcing via a further pathway: in contrary to wet-deposition, dry deposition velocities (DV) greatly depend on the land-use type which determines the resistance of the surface and canopy layer (Zhang et al., 2003; Cherin et al., 2015; Hardacre et al., 2021). In urban environments, vegetation is greatly reduced (expressed for example in term of the leaf-area-index LAI reduction). As plants represent a major sink for many gaseous air pollutants (via stomatal uptake), it is clear that over urban areas, this sink is missing or is strongly reduced. For example, based on the later study, over urban land-surface the typical DVs of nitrogen dioxide ($NO_2$) and ozone ($O_3$) are about half of that above agricultural land like crop (as typical rural land-surface type). Nowak and Dwyer (2007) calculated a net average pollutant removal if additional trees were planted in select US urban areas. Mcdonald-Buller et al. (2001) also showed that ozone and $NO_2$ is greater in a photochemical model if the landuse information supplied contains higher fraction of urban landuse type. In general it seems that besides other effects, urbanization leads to increased ozone concentrations due to reduced deposition values too (Song et al., 2008; Tao et al., 2015) while dry-deposition itself is an important factor determining ozone pollution (Galmarini et al., 2021). However, for some secondary pollutants like $HNO_3$, $H_2SO_4$, $H_2O_2$, HONO or $NH_3$, the removal in case of wet canopies is higher for urban areas than for rural ones (e.g. crops) due to their high solubility and reactivity with solid surfaces (Zhang et al., 2003). Urbanization in case of these species means higher DVs leading to decrease of their concentrations. It is thus clear, that the final air-pollution caused by urbanization has another contribution represented by the modified (increases and decreases too) dry-deposition uptake potential of urban land-surface compared to rural/natural one.

Finally, vegetation does not act only as a sink of pollutants via dry-deposition by it also emits large amounts of biogenic hydrocarbons (biogenic volatile organic compounds; BVOC; Kesselmeier et al. (1999)). Due to their reactivity and potential to form peroxy-radicals, they contribute to the formation of tropospheric ozone (Situ et al., 2013; Tagaris et al., 2014). As

mentioned already above, during urbanization, the vegetation is strongly reduced which will result in a decrease of BVOC emissions. Song et al. (2008) for example showed up to 10% reductions due to urbanization in Texas. As urban areas are usually VOC-limited environment, reduced BVOC emissions is expected to lead to reduced ozone concentrations (Song et al., 2008). It has to be noted that anthropogenic emissions from urban areas encompass the emissions of VOC compounds of typically biogenic origin (like isoprene and monoterpenes, (Wagner et al., 2014; Panopoulou et al., 2020)), probably these emissions are much smaller than other VOC emissions (Guo e a., 2022) and cannot overweight the reduction due to reduced vegetation.

The urbanization induced BVOC emission modifications have a further sub-component acting via the modified meteorological conditions in cities. Indeed, urban temperatures are higher than rural ones and there is an indication that urban cloudiness, at least for European cities is slightly reduced too (Karlický et al., 2020). These effects have direct impact on the biochemistry of plants and thus on the amount of emitted BVOC as higher temperatures and more solar radiation promote these emissions (Guenther et al., 2006). This means that due to urbanization, BVOC emissions are suppressed by reducing the vegetation fraction, however, more favorable weather conditions act in opposite way making these effects counteracting. Although there is an indication that the former (vegetation) effect is dominant (Li et al., 2019).

In summary, urbanization substantially affects air-quality while the final urban pollutant concentration levels are a result of multiple impacts that add to the background (i.e. that without urbanization) air pollution:

1. The effect of urban emissions ("DEMIS")

2. The effect of the urban canopy meteorological forcing (UCMF) on pollutant transport and chemistry ("DMET")

3. The effect of modified dry-deposition associated with modified land-cover ("DLU_D")

4. The effect of modified emissions of biogenic volatile compounds (BVOC) due to modified land-cover (i) and meteorology (ii) ("DBVOC").

As seen above, many studies looked at the total impact of urbanization or at some of the individual contributors listed. However, they did not systematically analyze the impact of each one of them. Here we propose a study to uncover the i) total impact of urbanization ("DTOT") and, more importantly, the contribution of ii) each of the urbanization related impacts (i.e. "DEMIS", "DMET", "DLU_D" and "DBVOC") over regional scale domain on the present day urban air pollution levels using coupled regional climate and chemistry transport models applied at moderate 9 km x 9 km horizontal resolution. To achieve this goal, we have to define the reference (or background; not to be confused with "background ozone" which is a well defined term) state to which these impacts will be gradually added: rural landuse without the effect of UCMF and only rural emissions (urban emissions removed, i.e. those falling within the city administrative boundaries) while present day landuse, emissions and climate is considered (2015-2016, see below). To evaluate the individual contributors to urbanization as well as their combined effect, we will gradually add each of them to the base simulation in a cascading fashion. To reduce the uncertainty of the results caused by the different geographical and climatic conditions of cities, we perform our analysis for a large ensemble of cities in central Europe: 19 cities in total. Although, a similar estimate across several urban areas was made in Huszar et al.

(2016a), they focused on the effect of emissions only (which corresponds to "DEMIS" in our study) while none of other effects (UCMF, effect of landuse on dry-deposition and effect of modified biogenic emissions) was considered. Further it is clear, that to some degree the urbanization of one geographic location will affect the air pollution of other urban area, however this effect was evaluated to be minor for emissions and UCMF if the cities analyzed are sufficiently far from each other (Huszar et al., 2014, 2016a). Therefor the selection of cities considers the requirement of sufficient distance from each other.

The study will focus on key gas-phase pollutants $NO_2$, $O_3$ and $SO_2$. $NO_2$ is one of the most important primary pollutants in urban environment responsible for reduced air-quality and being a precursor for secondary pollutants like ozone or inorganic fine aerosol (Im and Kanakidou, 2012; Stock et al., 2013; Mertens et al., 2020). Ozone is formed in urban plumes when NOx and VOCs mix together promoted by solar radiation (Xue et al., 2014). Finally, sulfur dioxide – a pollutant originating mainly from fossil fuel combustion in energy production (Guttikunda et al., 2003) – has although undergone significant reduction in European cities during the last decades, it remains of concern, especially in eastern European countries (e.g. in Poland; (EEA, 2019)).

Of course, the urban air-pollution is affected not only by local effects. Emissions from other areas (rural or other, even distant cities) constitue a major fraction of urban air pollution (Im and Kanakidou, 2012; Huszar et al., 2016a). Further the background regional air pollution is an important factor that play role e.g. in urban ozone burden (Yan et al., 2021). Also the UCMF can have regional effects and the UCMF due to one city can have effect on other one (Huszar et al., 2014). However, in this study we are interested in the local effects, i.e. the effect of rural-urban-transformation on the local final air-pollution and concerned about the effect of the background atmosphere or the effect from other urban areas.

The study is structured as follows: after the Introduction, the experimental tools (models), their configuration and the data used are presented. Next, the experiments performed are presented followed by the Result section and finally, these are discussed and conclusions are drawn.

## 2 Methodology

### 2.1 Models used

The study is based on numerical experiments carried out using regional climate models (RCM) coupled to a chemistry transport model (CTM). To describe the regional climate, two RCMS as meteorological driver are used: RegCM version 4.7 and WRF version 4.0.3. Chemistry was resolved with the chemical transport model CAMx in version 7.10. The decision behind choosing two regional meteorological drivers is to achieve, at least to some degree, more robust results given the fact that the modelled meteorological conditions over cities greatly impacts the chemical concentrations (Ďoubalová et al., 2020; Huszar et al., 2018b).

As the models used and the parameterizations applied are almost identical to those in Huszar et al. (2021), here we list the most important details. RegCM4.7 is a regional scale climate model with both hydrostatic and non-hydrostatic dynamics (Giorgi et al., 2012). The schemes adopted are: Tiedtke scheme (Tiedtke et al., 1989) for convection, Holtslag scheme (HOL; Holtslag et al., 1990) for PBL parameterization and the 5-class WSM5 moisture scheme (Hong et al., 2004) for microphysics.

The atmosphere-biosphere-surface exchange in RegCM was calculated using the Community Land Model (CLM) version 4.5 (Oleson et al., 2013) land-surface scheme and to resolve the urban scale meteorological phenomena the CLMU module within CLM4.5 is invoked (Oleson et al., 2008, 2010). CLMU considers the traditional canyon geometry approach meaning that cities are represented as networks of street-canyons with specified geometrical and surface parameters (Oke et al., 2017).

WRF (Weather Research and Forecasting Model) is a regional weather prediction and climate model with detailed description provided by Skamarock et al. (2019). In our modelling setup, the Grell 3D convection scheme (Grell, 1993), the BouLac PBL scheme (Bougeault and Lacarrère, 1989), and the Purdue Lin scheme (Chen and Sun, 2002, PLIN;) for microphysics were used. The urban canopy meteorological effects were resolved using the Single-Layer Urban Canopy Model (SLUCM; (Kusaka et al., 2001)).

For the chemistry simulations the chemistry transport model CAMx version 7.10 (ENVIRON, 2020) was used (i.e. we used the most up-to-date version for CAMx available). CAMx is an Eulerian photochemical CTM implementing multiple gas phase chemistry schemes (Carbon Bond 5 and 6, SAPRC07TC etc.) with the Carbon Bond 6 revision 5 (CB6r5) scheme used in this study. CB6r5 includes updates to chemical reaction data from IUPAC (IUPAC, 2019) and NASA (Burkholder et al., 2019) for inorganic and simple organic species important for the formation of ozone. Apart from the inclusion of CB6r5 mechanism, this version of CAMx includes important modifications of secondary aerosol formation via oxidation of VOCs which can have feedbacks on the total VOC and thus ozone concentrations. To complete the atmospheric chemistry with aerosol physics, static two mode approach was considered. The ISORROPIA thermodynamic equilibrium model (Nenes and Pandis, 1998) was invoked for the secondary inorganic aerosol formation. Secondary organic aerosol (SOA) were partitioned from their gas-phase precursors using the SOAP equilibrium scheme (Strader et al., 1999). For wet deposition, the Seinfeld and Pandis (1998) scheme is used while dry-deposition is treated using the Zhang et al. (2003) method. The Zhang method incorporates a 3-resistance equation for deposition velocity (DV) incorporating the aerodynamic resistance ($r_a$), the above-canopy quasi-laminar sublayer resistance ($r_b$) and the overall canopy resistance ($r_c$) while DV is calculated as the reciprocal of the sum of these resistances. An important component of $r_c$ is the resistance represented by vegetation with the so called in-canopy aerodynamic, the stomatal, the mesophyll and the cuticle resistances. Over urban surfaces or any other non-vegetated surfaces these are not defined however to use the same equations for such landuse categories in the dry-deposition model, very large values are applied (e.g. $10^{25} sm^{-1}$). Regarding the aerodynamic resistance representing the bulk transport trough the lowest model layer via turbulent diffusion, its magnitude depends on the intensity of turbulence, which in turn depends on wind speed, surface roughness, near-surface temperature lapse rate and solar insolation. Over urban areas these are strongly modulated implying strong influence also on the deposition velocities. Finally, the quasi-laminar sublayer (or boundary) resistance $r_b$ represents molecular diffusion through the thin layer of air directly in contact with the surface and this is assumed to be the function of the molecular diffusivity of each pollutant regardless the surface it is deposited on. In this dry-deposition model, the aerodynamic resistance has a strong dependence on the temperature via decreased stability near the surface and thus more efficient turbulent diffusion towards the surface (Louis, 1979). Also the stomatal resistance decreases with higher temperatures due to wider stomatas (Zhang et al., 2002); this holds up to a threshold maximum temperature at which stomatas suddenly

close. These temperatures are usually however not reached in the climate of the region. Therefore increased dry-deposition velocities are expected as the result of increased temperatures in urban areas.

Meteorological preprocessor is used to convert the RegCM and WRF meteorological data into model-ready driving data for CAMx: for the WRF, it was the wrfcamx preprocessor which is provided along with the CAMx code http://www.camx.com/download/support-software.aspx while for RegCM, the RegCM2CAMx interface was applied (Huszar et al., 2012). The vertical eddy diffusion coefficients ($K_v$) are diagnosed from the available meteorological data on RegCM and WRF output using the CMAQ diagnostic approach (Byun, 1999). Temperature, pressure, humidity, cloud/rain water content are defined at cell

centers along with pollutant concentration, and CAMx considers them as grid cell average conditions. On the other hand, wind and diffusion coefficients are carried at cell interfaces to describe the mass transfer across each cell face. Coupling between CAMx and the driving models is offline, which implies that no feedbacks of the pollutant concentrations on WRF/RegCM radiation and microphysical processes were considered. Indeed, Huszar et al. (2016b) showed on 10yr long simulations that the long term radiative effects of urban pollutant emissions and secondarily formed pollutants (like ozone) is rather small which

justified this choice.

## 2.2   Model setup and data

Model simulations were performed over identical domains (parent and nested ones) and for identical period as in Huszar et al. (2021), i.e. years 2015-2016 with 9 km, 3 km and 1 km horizontal resolution centered over the Czech capital, Prague (50.075° N, 14.44° E; Lambert Conic Conformal projection). In vertical, the model grid has 40 layers in both meteorological

driving models. The thickness of the lowermost layer is about 30 m and the top of the model's atmosphere reaches 5 hPa (about 36 km). The simulated time-period is 2014 Dec – 2016 Dec (the first month used as spin-up). Tie et al. (2010) argued that the ratio of diameter of the analyzed city to model resolution should be at least 6:1, which means that in our case, 6 km or smaller horizontal grid step should be used to resolve the impact of urbanization for the cities chosen (see below). For Prague, which is modelled at 1 km, this is fulfilled. Other cities outside the inner 1 km nested domain are treated at coarser resolution but many

studies found that the impact of the resolution of emissions and models on urban species concentrations is rather small: e.g. Hodneborg et al. (2011) showed that coarse resolution can lead to higher ozone modelled by around 10% while Markakis et al. (2015) found only moderate sensitivity (8%) on model resolution. Similarly, Wang et al. (2021a) showed that ozone production is reduced when high resolution is applied but the reduction is only about 8% for ozone.

    The ERA-interim reanalysis (Simmons et al., 2010) is used as forcing data. The 3 and 1 km domains are then driven by

220 the corresponding parent domains with one-way nesting. Chemical boundary conditions are based on the CAM-chem global model data (Buchholz et al., 2019; Emmons et al., 2020). Landuse data (for both climate models and for the dry-deposition scheme in CAMx) was derived from the high resolution (100 m) CORINE CLC 2012 landcover data (https://land.copernicus.eu/pan-european/corine-land-cover, last access Aug 8, 2022) as well as from the United States Geological Survey (USGS) database for gridcells with no information from CORINE. In RegCM, fractional landuse is considered while in WRF, each

225 gridcell is attributed the dominant landuse, which brings some accounting for the uncertainty related to the urban land-cover

representation. This means that due to the fact that the landuse is represented differently in WRF and RegCM, partly urbanized surfaces and their effects can be differently accounted for in the two models.

The European CAMS (Copernicus Atmosphere Monitoring Service) version CAMS-REG-APv1.1 inventory (Regional Atmospheric Pollutants; (Granier et al., 2019)) for year 2015 was used as anthropogenic emission data for areas outside Czech Republic. There, high resolution national data were adopted: the Register of Emissions and Air Pollution Sources (REZZO) dataset issued by the Czech Hydrometeorological Institute (www.chmi.cz) and the ATEM Traffic Emissions dataset provided by ATEM (Ateliér ekologických modelů – Studio of ecological models; www.atem.cz) were used. These data provide activity based (SNAP – Selected Nomenclature for sources of Air Pollution) annual emission totals of oxides of nitrogen ($NO_x$), volatile organic compounds (VOC), sulfur dioxide ($SO_2$), carbon monoxide (CO), PM2.5 and PM10 (particles with diameter less than 2.5 and 10 $\mu$m) and ammonia ($NH_3$). CAMS data are defined on a regular Cartesian lat–lon grid while the Czech datasets are provided as area, line (for road transportation) or point sources (in case of area sources these are usually irregular shapes corresponding to counties with resolution from a few 10 m to 1-2 km). The Flexible Universal Processor for Modeling Emissions (FUME) emission model (http://fume-ep.org/; Benešová et al., 2018) is used to preprocess the mentioned emission inventories to CTM-ready emission files, including preprocessing the raw input files, the spatial remapping of the data into the model grid, chemical speciation and time-disaggregation from annual to hourly emissions. Speciation factors and time-dissaggregation profiles were taken from Passant (2002) and van der Gon et al. (2011), respectively. The temporal factors contain activity sector specific monthly, weekly and hourly factors used to decompose the annual totals into hourly emissions. Geographic dependence is not considered here, however, the timezone information and the associated time shifts are accounted for.

Emissions of biogenic origin are calculated offline using the MEGANv2.1 (Model of Emissions of Gases and Aerosols from Nature version 2.1) model (Guenther et al., 2012) based on RegCM and WRF meteorology (temperature, short-ware radiation, humidity, soil moisture). The necessary input for MEGAN including leaf-area index data (its annual cycle), plant functional types and emission potentials of different plant types are not part of the CORINE landuse data and were derived independently from Sindelarova et al. (2014, 2022). It has to be mentioned here that along with the calculation of biogenic VOC data, MEGAN also calculates the fluxes of soil-biogenic NO (nitrogen monoxide) emissions as a result of bacterial activity in soil according to (Yienger and Levy, 1995). As these emissions are a function of LAI and meteorological conditions, part of the "DBVOC" impact will be composed of soil-NOx emissions modifications. Not presented here, in our experiments the soil-NOx emissions are about two orders of magnitude smaller compared to the BVOC emissions and their effect is expected to be much smaller including the effect of their urbanization induced modifications. It has to be also stressed that BVOC emissions are strongly temperature dependent while higher temperatures trigger stronger emissions. In this regard, urbanization induced temperature enhancement is expected to lead to stronger BVOC fluxes. Wildfire emissions can be potentially important episodically and can significantly contribute to levels of gaseous pollutants like NOx and CO and to improve overall model performance (Lazaridis et al., 2008), they are however significant mainly over southern Europe and Mediterranean and not over our focus area (central Europe). Moreover, wildfire emissions normally do not occur in urban areas so do not contribute to the impact of urbanization.

A key task was to isolate the emissions originating from urban areas (see further for details about the chosen cities). In this regard, urban areas were identified based on the administrative boundaries of chosen cities. We used the GADM public database (https://gadm.org) for their definition. While masking of inventory emissions based on the GADM shapes corresponding to cities, it had to be ensured that the partition between the "city" and "non-city" parts of emission segments (inventory gridcells or irregular shapes in case of Czech emissions) that lie over the city boundary is correctly calculated. For this purpose, the
masking capability of FUME was adopted.

The cities chosen in the analysis are Berlin, Brussels, Budapest, Cluj-Napoca, Cologne, Frankfurt, Hamburg, Krakow, Lodz, Lyon, Milan, Munich, Prague, Torino, Vienna, Warsaw, Wroclaw, Zagreb, Zurich. They are also highlighted in Fig. 1 including the 9 km domain terrain elevation. The choice of the cities regarded the same criteria as in Huszar et al. (2021): the size of the city comparable to one 9 km $\times$ 9 km gridcell, sufficient distance between cities to eliminate inter-city influences, minimal
orographic variability to reduce orographic effects (Ganbat et al., 2015), no coastal cities to eliminate the effect of asymmetric landuse, like e.g. the sea-breeze effect (Ribeiro et al., 2018). Although strict emission control policies, these cities are still often burdened with high air pollution for pollutants as $NO_2$ and $O_3$ (EEA, 2019; Khomenko et al., 2021; Sokhi et al., 2022).

## 2.3    Model simulations

The study intends to evaluate the urbanization impact on air quality while we attempted to decompose the total impact into
individual contributors listed in the Introduction. This requires to perform a series of model experiments with individual effects added gradually one-by-one to reference state to end up with the real situation corresponding to full urbanization. In Huszar et al. (2018a, b) we performed similar decomposition for the urban induced meteorological effects (i.e. the UCMF) and their impact on air-quality. Here we adopt this approach, but it will not concern the UCMF solely as in the mentioned works but the entire impact of urbanization while UCMF will be treated as one effect.

The simulations performed are summarized in the Tab. 1 and 2 for RCM and the underlying CTM simulations, respectively. A pair of simulation was performed with both RegCM and WRF with ("Urban") and without ("Nourban") considering urban land-surface. In the latter case, landuse was replaced by "crops" as the most common rural landuse type in the region analyzed. While all RegCM simulations and CAMx simulations driven by RegCM were performed on nested domains (9 km, 3 km and 1 km), the WRF simulations and CAMx ones driven by WRF were done only over the parent 9 km domain as WRF served as
a complementary model to account for the uncertainty in the driving meteorology, especially with regard to UCMF; note that the urban canopy model is different in WRF than in RegCM.

As for the CAMx simulations, they differ based on the inclusion of urbanized/rural land-surface, the UCMF (acting on both atmospheric chemistry in general and on BVOC fluxes) and the urban emissions. In this regard, we performed 6 experiments summarized in Tab. 2. The reference experiment called "ENNrrN" represents the hypothetical background state
without urban emissions and with the urban land-surface replaced by rural land-surface in RCMs and CTM as well as in the BVOC model (MEGAN). The reference simulation is not to be confused with preindustial state: we assumed current climate (GHG concentration) and current background large scale chemical concentrations. In the next experiment, "ENYrrN", only the urban emissions are considered (turned on). In the 3rd experiment, "ENYurN", the urban landuse was "turned-on" for the

dry-deposition in CAMx. In the 4th experiment, "ENYuuN", the urban landuse is "turned-on" also for the biogenic emission model. In the 5th experiment, "ENYuuU", both the urban landuse and the UCMF is accounted for the biogenic emissions model and finally, in the 6th experiment, "EUYuuU", all the urbanization-related effects are considered, representing the most realistic case.

In the first experiment where urban emissions are disregarded, we removed urban emissions only for the 19 cities chosen for the analysis. For the effect of rural-urban landuse transformation on meteorological conditions, dry-deposition and biogenic emissions, we replaced the urban land by rural one over the entire domain (i.e. not only for the cities chosen). It is clear that this has effect on the background level of air pollutants not only local urban levels, but the effect is probably much smaller than local effects as 1) emissions from these areas were still considered, 2) the urban meteorological effects from these (minor) urban areas have rather small influence or air pollutants as the UCMF is also small (see e.g. Huszar et al. (2014)). In Fig. 1, we plotted the model orography and the analysed cities as red squares. For urban landuse information used in our study, please refer to Karlický et al. (2020) (Fig. 1 and 2), who used identical landuse as in our study.

Mathematically, with respect to the rural-urban transformation (RUT), the concentration $c_i$ of a pollutant $i$ for a chosen city is given by:

$$c_i = c_{i,rural} + \Delta c_{i,RUT}, \tag{1}$$

where $c_{i,rural}$ is the average concentration before RUT and $\Delta c_{i,RUT}$ is the total impact of urbanization.

In this study, we are concerned about the contributors to $\Delta c_{i,RUT}$ (regardless of their sign).

$$\Delta c_{i,RUT} = \Delta c_{i,EMIS} + \Delta c_{i,MET} + \Delta c_{i,LU_D} + \Delta c_{i,BVOC}, \tag{2}$$

where $\Delta c_{i,EMIS}$, $\Delta c_{i,MET}$, $\Delta c_{i,LU_D}$ and $\Delta c_{i,BVOC}$ are the impacts of urban emissions, the impact of the urban canopy meteorological forcing, the impact of modified landuse on dry-deposition and the impact of modifications of BVOC emissions, denoted above as "DEMIS", "DMET", "DLU_D" and "DBVOC". The $\Delta c_{i,BVOC}$ impact can further be decomposed into the part caused by modified land-cover (reduced vegetation in terms of changes in leaf-area-index LAI, "DBVOC_L") and modified meteorological conditions ("DBVOC_M"):

$$\Delta c_{i,BVOC} = \Delta c_{i,BVOC_L} + \Delta c_{i,BVOC_M}. \tag{3}$$

These impacts will be calculated from the experiments listed in Tab. 2 in the following way (the experiment number is shown in paranthesis):

$$\Delta c_{i,RUT} = EUYuuU(6) - ENNrrN(1) \tag{4}$$
$$\Delta c_{i,EMIS} = ENYrrN(2) - ENNrrN(1) \tag{5}$$
$$\Delta c_{i,MET} = EUYuuU(6) - ENYuuU(5) \tag{6}$$
$$\Delta c_{i,LU_D} = ENYurN(3) - ENYrrN(2) \tag{7}$$
$$\Delta c_{i,BVOC} = ENYuuU(5) - ENYurN(3) \tag{8}$$
$$\tag{9}$$

It has to be noted, that in reality these effects act simultaneously and feedback are present between them so their effects are not additive. The way how we calculated the individual impacts (contributors) however allow us to consider them to be additive, i.e. their sum is the total impact of urbanization.

## 3 Results

### 3.1 Validation

This model configuration (same input data, same domain) underwent a detailed validation including both meteorology and air-quality in Huszar et al. (2020b) and Huszar et al. (2021). However, due to the fact, that in our CAMx simulations, instead version 6.50 a newer version 7.10 was used and instead of the CB5 we used the newer CB6 chemistry mechanism, we provide a brief account for validation. For comparison with observations, the AirBase European air quality data (http://www.eea.europa.eu/data-and-maps/data/aqereporting-1) for the modelled years was used, while all urban and suburban type background stations (AirQualityStationArea = "urban"/"suburban" and AirQualityStationType = "background") were used from a subset of the analyzed cities. This sub-selection considered the largest cities from the total of 19 ones where sufficient number of stations were available.

In Fig. 2, the comparison of average monthly means of modelled and measured concentrations of the three analyzed pollutant is shown while the full experiment ("EUYuuU") was used which represents the real case. For ozone, we also included the average summer diurnal cycle, as daily peak values are more important for this pollutant than the averages values. For Prague, results are taken from the 1 km nested domain, otherwise they are extracted from the 9 km regional domain. For $NO_2$, there is a generally acceptable match between the model and observations with model biases up to $10\,\mu gm^{-3}$. While concentrations from Jan to Apr are usually underestimated, during summer, CAMx generates a positive bias, except Berlin, where there is an underestimation of $NO_2$ from March to September and an overestimation during the rest of the year. During late autumn the model bias is usually negative with large differences between cities. The WRF driven CAMx concentrations are usually lower then for RegCM/CAMx during summer which means higher model bias (up to $-10\,\mu gm^{-3}$). During winter, WRF/CAMx gives higher concentrations than RegCM/CAMx resulting in smaller bias.

For $O_3$, both RegCM/CAMx and WRF/CAMx well capture the annual cycle with some overestimation of concentrations during late spring (by about $10\text{-}20\,\mu gm^{-3}$) and an underestimation during late summer (by a similar magnitude) and in general, these two simulations are very similar. During winter, there is a small negative (up to $-10\,\mu gm^{-3}$) model bias present. The summer diurnal cycles show a good agreement in the basic measured pattern, including the timing of the maximum. The maximum values are sometimes underestimated (by up to $20\,\mu gm^{-3}$ for some cities) especially for the RegCM driven runs. In general, WRF meteorology causes higher simulated maximum ozone. During night-time, ozone is often overestimated by around $10\text{-}20\,\mu gm^{-3}$ and clearly, the WRF driven run performs better during this part of the day.

Regarding $SO_2$, the model fails to capture well the annual cycle. During winter, both models usually underestimate the concentrations up to $1\text{-}2\,\mu gm^{-3}$ except Budapest and Berlin, where overestimation occurs in model. During summer, measured

SO$_2$ concentrations are usually smaller and the models somewhat reflects this fact, but still large biases are present and the models are unable to capture correctly the annual cycle for some cities (e.g. Budapest and Vienna).

## 3.2 The overall impact of individual contributors to RUT

Firstly, we evaluated the impact of individual contributors to the RUT as well as the total impact in terms of 2015-2016 DJF and JJA averages (in case of ozone as summer average only), averaged across the chosen cities and the model ensemble (i.e. average of the RegCM and WRF driven runs). Values are taken from gridbox covering the center of a particular city. The results are shown in Fig. 3 as boxplots showing the 1st and 3rd quartiles as well as the median values and the minimum and maximum.

The analysis showed (as expected) that from the four contributors to RUT, two are much stronger than the other two. Therefor in the plots, we separated them from the minors ones (including the total impact).

For all three gas-phase pollutants, the impact of emissions ("DEMIS") is the largest in magnitude in both seasons. For NO$_2$ it ranges (i.e. the 25% to 75% percentile) from 4 to about 8 ppbv and from 5 to 10 ppbv for JJA and DJF, respectively. For SO$_2$, the numbers are somewhat smaller as cities, at least in the region in focus, are not so strong SO$_2$ emitters (compared to

NO$_2$): an increase by 0.4 to 1.5, and 0.8 to 1.6 ppbv for JJA and DJF, respectively, is seen. For O$_3$, the impact on the summer maximum daily 8-hour average concentration (MDA8) is characterized by decrease due to titration (as expected) by 3 to 6 ppbv.

The impact of the urban canopy meteorological forcing ("DMET") is characterized by a decrease for the primary pollutants: for NO$_2$, the decrease is usually between 1 and 6 ppbv for JJA and between 1 and 3 ppbv in DJF with the maximum surpassing

zero meaning that in some cities, a slight increase was modelled. In case of SO$_2$, the impact of UCMF is smaller, up to 0.6 ppbv and 0.4 ppbv decrease in JJA and DJF, respectively. For O$_3$, the impact is an increase about 2 ppbv.

In case of minor contributors, the impact of BVOC is considerable for ozone only and as expected, for the other pollutants it acts as a minor modulator of the overall chemistry (e.g. influencing the hydroxyl budget) therefor having a very small impact. In case of NO$_2$, the impact is a slight increase by around 0.01 ppbv in JJA and negligible in winter. For SO$_2$, it is near zero in

both seasons. For ozone, which is directly influenced by biogenic emissions, the impact is a decrease by around 0.4 to 1 ppbv as JJA average. The impact of modified dry-deposition due to urbanized landuse is characterized by an increase (0.02 to 0.08 ppbv) for NO$_2$ in JJA and an opposite impact in winter (around 0.01 to 0.04 ppbv decrease). For SO$_2$, the impact in summer can be both negative and positive (from -0.01 to 0.01 ppbv) with a near zero average. In winter, there is a decrease by about 0.02 to 0.07 ppbv. For ozone, the impact of landuse change is an increase between 1 and 2 ppbv.

Finally, the total impact is an increase for all pollutants and quantities: for NO$_2$ it is about 1-5 ppbv in JJA and 4-8 ppbv in DJF, for SO$_2$ it ranges from 0 to 1 ppbv in JJA and about 0.5 to 1 ppbv in DJF. For JJA MDA8 ozone, the total impact is characterized by an increase up to 2 ppbv.

## 3.3 The spatial distribution of the impacts

The boxplots presented above give an overview of the averaged impact across all the cities including the distribution around the

median value. To obtain a spatially resolved information, we plotted here also the 2-D distribution of the individual contributors.

### 3.3.1 The impact of urban emissions (DEMIS)

In Fig. 4 the DJF and JJA average spatial impact of urban emissions ("DEMIS") on the near-surface concentrations of $NO_2$, $SO_2$ and $O_3$ is shown.

In case of $NO_2$, the impacts reaches 4-6 ppbv in the core of the cities and remains high over surrounding areas (up to 0.5 ppbv over large areas in DJF, especially in WRF driven simulations). In summer, the spatial extent of the emission impact is smaller getting below 0.1 ppbv over rural areas. The result from Prague at high resolution reveals that the high emission impact is concentrated to the very center of the city (reaching 4-6 ppbv).

For $SO_2$, there is a larger spread between cities with large contributions over Poland reaching 6 ppbv (in both seasons), while for other cities, the contribution is smaller, up to 2-3 ppbv. The contribution over rural areas is large in Poland (up to 0.2 ppbv) but remains below 0.1 ppbv in other regions. The impact of emissions over Prague reaches 1 ppbv in DJF with contributions up to 0.1 ppbv in its vicinity. In summer, due to low emissions ($SO_2$ is emitted largely by heating) the contributions are very small, reaching 0.5 ppbv in some hotspots within the city.

Ozone is usually titrated in city centres which corresponds to the impact of emission on its concentrations. They decreased over cities by up to 3-4 ppbv, while further from cities, where urban NOx mix with rural emissions, ozone increase occurs up to 1 ppbv as MDA8. Over Prague, the decrease is limited to the city area. Over its vicinity, the impact becomes positive with 0.5-1 ppbv increase (similar as seen for other cities).

### 3.3.2 The impact of modified meteorological conditions (DMET)

Fig. 5 presents the DJF and JJA average spatial impact of the urban canopy meteorological forcing ("DMET") on the near-surface concentrations of $NO_2$, $SO_2$ and $O_3$.

In case of $NO_2$, while for RegCM/CAMx the impact is characterized usually by decrease by 1-2 ppbv with some urban areas showing even an increase (up to 2 ppbv), for WRF/CAMx a clear decrease occurs up to 3 ppbv. For Prague, the highest decreases are modelled in the city center reaching about 3 ppbv in summer and about 2 ppbv in winter. In general, the winter impact is comparable to summer one (slightly stronger for WRF/CAMx).

For $SO_2$ the impact is weaker and constitutes both decreases (in cities) and increases (over their vicinity) with changes in the range -3 to 3 ppbv. In case of WRF/CAMx simulations, the impact is more straightforward with the decrease dominating reaching 3 ppbv in both seasons.

Finally, summer MDA8 ozone increases due to UCMF up to 2-3 ppbv over cities while over rural areas, a slight decrease is modelled up to 1 ppbv appearing in the RegCM/CAMx simulation. Over Prague, the largest increases are modelled in the city center reaching 2-3 ppbv.

### 3.3.3 The impact of dry-deposition modifications (DLUC_D)

The impact of the urban land-cover via modified dry-deposition ("DLU_D") is plotted on Fig. 6. In general, the impacts are much smaller for $NO_2$ and $SO_2$ than seen for the emission or the UCMF impact for these pollutants above. For $NO_2$, the

DJF and JJA impacts differ in sign (in accordance with the boxplots seen in Fig. 3) and the spatial distribution is somewhat different in WRF/CAMx than in RegCM/CAMx. In DJF, $NO_2$ concentrations decreased over cities by up to 0.04 ppbv, with

some higher decreases over Italy (Milan) up to 0.1 ppbv. In the WRF driven experiment, some increases over the Benelux states are also seen reaching 0.06 ppbv. For Prague, the decrease is maximal in the city center reaching 0.04 ppbv. During JJA, the "DLU_D" impact is positive reaching 0.1 ppbv in both models with some slight decreases around Milan. Over Prague, the increase is even stronger and exceeds 0.05 ppbv.

For $SO_2$, there are clear decreases modelled during DJF reaching 0.1-0.2 ppbv over city centres. The impacts are slightly

stronger in WRF driven CAMx runs and are about -0.03 ppbv over Prague's center. During JJA, the $SO_2$ response is very small and positive in the RegCM/CAMx experiments up to 0.1 ppbv increase in some city centres, specially over eastern Europe were $SO_2$ emissions are higher. Decreases similar to the DJF impact remained in the WRF/CAMx simulation. Over Prague, almost zero impact is modelled (lying between -0.01 and 0.01 ppbv with some positive impact around strong point sources north from the city).

A much stronger response to changes in dry-deposition is modelled for summer $O_3$ with a clear increase reaching 2 ppbv in city centres and being high over rural areas too (up to 1 ppbv increase). Over Prague, the increase is usually between 1.5-2 ppbv exceeding 2 ppbv in the very core of the city.

To facilitate the interpretation of the simulated responses of concentrations to "DLU_D", we also mapped the geographical distribution of the "DLU_D" impact on the deposition velocities (DV is standardly provided on CAMx output), seen in Fig. 7

taken from the RegCM driven CAMx simulations (and not showing the Prague 1 km case). For $NO_2$, dry deposition velocities decreased by around 0.2 $mm.s^{-1}$ in DJF and a stronger decrease, reaching -0.6 $mm.s^{-1}$ in city centres is modelled in JJA. For $SO_2$, the DJF and JJA maps differ in sign. For winter, deposition velocities increased in cities by up to 0.4-0.6 $mm.s^{-1}$ while during summer, similar decreases are simulated compared to $NO_2$ (around -0.4 to -0.6 $mm.s^{-1}$). For $O_3$, both seasons are characterized by decreases: by around 0.2 $mm.s^{-1}$ in DJF and with a stronger decreases in city centres in JJA, reaching

445    -1.5 $mm.s^{-1}$. The WRF/CAMx impacts are very similar and are not shown here.

### 3.3.4   The impact of biogenic emissions (DBVOC)

The urbanization induced changes in BVOC emissions (via reduced vegetation cover and modified temperatures; "DBVOC") and their consequent effect on summer ozone and $NO_2$ concentrations are plotted in Fig. 8. As BVOC emissions are of minor importance in winter and the effect on $SO_2$ are almost zero, we show only the summer impacts for these two pollutants.

For $NO_2$, the "DBVOC" impact results in increases usually up to 0.06 ppbv while much stronger increases are modelled over northern Italy (around Milan) around 0.1 ppbv in both RegCM and WRF driven simulations. For Prague, the maximum increase is between 0.2-0.3 ppbv.

For $O_3$, decreases are modelled reaching -1 ppbv over many cities and reaching -0.2 to -0.5 ppbv over rural areas (in the RegCM driven experiment). A stronger decrease is modelled (again) over northern Italy up to -2 ppbv over Milan. Prague is

characterized by a decrease usually between -0.5 and -1 ppbv.

The above presented impacts are the result of modified BVOC emissions, therefor we also plotted the summer changes of isoprene (ISOP) as a major component of such emissions. As changes of these emissions is the result of two components constituted of vegetation change via LAI change ("DBVOC_L") and modification of meteorological conditions (the UCMF; denoted "DBVOC_M" in the introduction), we plotted the two contributors separately in Fig. 9 as absolute and relative change. We were also interested whether the reference with respect to which the change is calculated matters. In others words, what is the difference between the "DBVOC_L" calculated at rural (NOURBAN, see Tab. 1) meteorology and "DBVOC_L" calculated at urban meteorology (URBAN in Tab. 1). Similarly for "DBVOC_M": it is calculated both with rural LAI and that adapted for urban conditions. The impact of vegetation change is an expected decrease in isoprene emissions by up to 15 $\mathrm{mol.km^{-2}hr^{-1}}$, with higher values over southern part of the domain, representing often a 80-90% decrease in relative numbers, especially for larger and dense urban areas like Milan (Italy). For smaller urban areas the decrease is around -5 to -20% (many of the gridcells are only partly covered by urban areas and so the emission decrease is correspondingly small). As seen from the figure, the changes calculated at rural and urban meteorological conditions are very similar (the case with urban meteorology is slightly higher). Regarding the isoprene emission modifications due to UCMF, they are usually much smaller (usually less than 0.05 $\mathrm{mol.km^{-2}hr^{-1}}$ or less than 0.5% in relative numbers). At some urban areas over Germany and over northern Italy and southern France, the change can reach 0.4 to 0.6 $\mathrm{mol.km^{-2}hr^{-1}}$ peaking at 1-2 $\mathrm{mol.km^{-2}hr^{-1}}$ over Italian urban areas, representing a 5-10% relative increase. The "DBVOC_M" is somewhat smaller if calculated at urban land-cover which is expected as the strongest meteorological modifications due to UCMF are over cities but in this case they affect a non-vegetated surface which implies a smaller effects. In summary, the BVOC emission changes associated with vegetation change are much more important than the modifications due to UCMF.

### 3.4 The diurnal variation of the impacts

Urban emissions have strong diurnal cycle caused by the typical cycle of human activities during the day. Moreover, the urban land-surface triggered meteorological modifications (UCMF) have also a strong diurnal pattern, e.g. temperature is impacted most during night, the wind impacts and turbulence modifications are the strongest during noon etc. (Huszar et al., 2018a, 2020a). Thus it is clear that the individual contributors to RUT analyzed here are expected to have also a diurnal cycle. Fig. 10 presents these cycles for the four contributors and three analyzed pollutants.

For $NO_2$ the diurnal pattern for the emissions impact ("DEMIS") follows the expected shape with two peaks during morning and evening rush hours reaching 10-12 ppbv and 9-11 ppbv in DJF and JJA, respectively. The diurnal cycle for the UCMF impact ("DMET") is negative throughout the whole day with peaking decrease during evening hours reaching -5 ppbv and -8 ppbv in DJF and JJA, respectively. In case of the impact of modified dry-deposition ("DLU_D") it has a somewhat different pattern in two seasons. In DJF, it is negative throughout the day with a strong peak during morning hours (-0.04 ppbv) and a smaller evening peak (-0.03 ppbv). In summer, this impact is positive almost during the whole day with two peaks during morning and early evening hours reaching 0.18-0.2 ppbv, while during night, the impact can be slightly negative up to -0.04 ppbv. The impact of BVOC changes ("DBVOC") is very small during winter with negative values peaking at less than -0.01 ppbv. During summer, the impact is stronger with a clear positive peak during evening hours reaching 0.06 ppbv.

In case of $SO_2$, the diurnal patter for the impact of emissions and UCMF is similar to $NO_2$. The emissions impact is peaking at morning and evening rush hours for JJA reaching 2.6-2.8 ppbv while in DJF, the maximum impact is reached at evening hours and the impact remains high during the whole night (around 2.5-3 ppbv). The "DMET" impact is negative with and evening peak reaching -0.06 and -0.03 in DJF and JJA, respectively. The impact on dry-deposition is negative in JJA with a maximum impacts during morning and evening hours reaching -0.05 to -0.07 ppbv. During JJA, the impact is positive during

495    day with increases up to 0.03 ppbv and decreases during night up to -0.04 ppbv. We already saw in the boxplots and also expected that the impact of BVOC emission change has an almost zero effect on $SO_2$, which is directly not tight chemically with VOC chemistry.

    Finally, for ozone, the impact of urban emissions is a decrease with two peaks during morning and evening hours reaching -10 to -12 ppbv in DJF and -8 to -10 ppbv in JJA. The impact of UCMF shows a clear increase peaking during evening hours

reaching around 5 and 10 ppbv during DJF and JJA, respectively. The impact of modifications of dry-deposition is positive throughout the day with a strong peak during noon to early evening hours - during DJF, the peaks reach 0.2 ppbv while a much stronger increase is modelled during summer reaching 1.5-2 ppbv. The impact of BVOC changes on ozone are virtually zero during DJF and are negative during JJA with a peak decrease around noon reaching 1 ppbv.

    We evaluated also the diurnal cycle of impact on deposition velocities, as this helps the interpretation of the "DLU_D". In

Fig. 11 the 2015-2016 winter and summer average of DV diurnal cycle for the three analyzed pollutants is plotted as well as the absolute values corresponding to the rural ("Nourban") case. In case of $NO_2$, DVs are reduced when turning rural landuse into urban one and the maximum decrease occurs during noon to early afternoon reaching -0.4 $mm.s^{-1}$ in DJF and a stronger decrease reaching -3 $mm.s^{-1}$ in JJA while during night, the change is close to zero. Similar decreases are calculated for ozone with somewhat smaller nocturnal values in DJF and a bit weaker decrease during summer peak values. For $SO_2$, the impact

on DV is different between DJF and JJA. During DJF, DV increases by 0.6 $mm.s^{-1}$ during night, while a smaller increase is calculated around noon time (0.2 $mm.s^{-1}$). During JJA, DV change for $SO_2$ is slightly above zero and a strong negative peak occurs during the day reaching about -1.5 to -2 $mm.s^{-1}$. Comparing with the absolute values, the impact of urban landuse change in winter can reach -10 to -20% for ozone and $NO_2$ while 20-30% for sulfur dioxide. In summer the decrease is even higher reaching 50% for ozone and $NO_2$ while for $SO_2$, the relative decrease is about 30-40%.

## 515    4   Discussion and conclusions

    We presented an analysis of the different contributors to the overall impact of urbanization (what we called here the rural-urban-transformation; RUT) on urban gas-phase air pollutant concentrations. We focused on the four most important contributors to RUT, namely the impact of urban emissions ("DEMIS"), the impact of the urban canopy meteorological forcing on pollutant chemistry and transport ("DMET"), the impact of modified dry-deposition due to the land-cover modifications ("DLU_D")

and the impact of modified biogenic emissions due to modified land-cover (and associated vegetation change) and modified meteorological conditions ("DBVOC"). By performing multiple simulations where each contributor of RUT was added one-

by-one to the reference state representing a land without urban land-cover and urban emissions, we could quantify them individually.

The validation showed a reasonable range of model bias and the annual cycles of pollutant concentrations for ozone and NO$_2$ were well captured. The same is true for the model ability to resolve the diurnal cycle of ozone. Regarding NO$_2$ biases, our results show a clear improvement from our previous study of Huszar et al. (2021) which used almost identical setup and the same emissions input. It is clear, that this improvement also cannot be explained by improved meteorology as the driving RegCM and CAMx simulations were the same. The only probably explanation is that the improvement were achieved by updating the chemistry mechanism in our simulations from CB5 to CB6r5. Indeed, CB6 was added to CAMx to take into account the long-lived organic compounds formed by peroxy radical reactions which serve as an inhibitor of OH recycling and reduces NOx removal by OH oxidation (Cao et al., 2021). Previously, Luecken et al. (2019) also found a better model performance for reactive nitrogen when CB6 was used instead of CB-V. The slight deviation of the monthly cycles from observed values is probably caused by not correct annual temporal disaggregation profiles which dependent only on the emission activity sector but not on the geographic location. Some studies using older chemistry mechanism also obtained larger NO$_2$ biases (Karlický et al., 2017; Tucella et al., 2012) which suggest that the accuracy of chemistry mechanism is probably very important. For ozone, monthly values were well represented by our model system and the choice of the chemistry mechanism probably contributed to this as in older studies using CB5 (or even the older CB-IV) (Zanis et al., 2011; Huszar et al., 2016a, 2020b) the biases were higher (moreover, the later study used the same emission data as here) and were often caused by strong night-time bias which seems to be partly removed in our study. Further, our results show a similar model-observation agreement than the large online-coupled model comparison study by Im et al. (2015). In case of SO$_2$, the model is rather unable to correctly resolve the annual cycle of near-surface concentrations. We saw this behavior in a similar manner in Huszar et al. (2016a) or in Karlický et al. (2017) too and points to deficiencies in the annual profile used to time-disaggregate annual emissions to monthly-ones. The SO$_2$ biases can be caused also by wrong vertical turbulent mixing as large quantities of this pollutant are emitted from tall stacks and they have to mixed down to the surface layer, which is greatly influenced by the model representation of vertical eddy-diffusivities. These are especially important in urban areas and still large uncertainty persists in their calculation (Huszar et al., 2020a). In summary, for NO$_2$ and O$_3$ we did not identify substantial model biases in simulating urban near-surface concentrations of the analyzed pollutants. For sulfur dioxide, our model failed to correctly resolve the annual cycle which suggest that the impact of urban SO$_2$ emissions can be also over/underestimated depending on the model bias and should be perceived with caution.

The total impact of urbanization on NO$_2$ was calculated to around 3(1÷5) ppbv in summer and 6(3÷8) ppbv in winter. These numbers are smaller than the annual mean contributions calculated for 2001-2010 in Huszar et al. (2016a) and higher contributions were also modelled by Im and Kanakidou (2012) however both simulated only the effect of urban emissions without considering the effect of the UCMF which decreases near-surface concentrations (see further). The total impact on SO$_2$ is between 0 and 1 ppbv in summer and 0.5-1.3 in winter, which is a smaller contribution than in Huszar et al. (2016a) due to much lower sulfur emissions in 2015 compared to the 2005 emissions used there and due to not considering the UCMF effects in the earlier study. The total average contribution for ozone summer MDA8 is about 1.5(0÷2) ppbv. In Huszar et al.

(2016a) for an ensemble of central European cities and Im et al. (2011a, b) for Mediterranean cities decrease of ozone was shown (and increase over rural areas, similar to our results), but they accounted for only the urban emission impact. Indeed, the urbanization via the UCMF increases ozone concentrations (Kim et al., 2015; Huszar et al., 2018a, 2020b) which can offset

the decrease seen solely due to urban emissions. For all three pollutants, the effect of emissions ("DEMIS") is stronger than the total effect of urbanization ("DTOT") due to the strong modulating effect of the urban canopy meteorological forcing. As already calculated by many (e.g. Wang et al., 2007, 2009; Struzewska and Kaminski, 2012; Zhu et al., 2015; Huszar et al., 2020a), the vertical eddy-diffusion is the most important component of UCMF which is strongly enhanced above urban areas. Consequently, it leads to reduced near-surface concentrations of primary pollutants (e.g. $NO_2$, $SO_2$) and an increase of ozone

due to reducing the titration by NO (Escudero et al., 2014; Xie et al., 2016a, b).

As for the impact of UCMF ("DMET") alone, our simulations showed a decrease by about 2 ppbv for $NO_2$, by 0.2-0.3 ppbv for $SO_2$ (for both seasons) and an increase of summer MDA8 ozone by about 2 ppbv. These numbers well fit previous findings in Sarrat et al. (2006); Struzewska and Kaminski (2012); Kim et al. (2015); Huszar et al. (2018a, 2020b). They concluded that three main components play the most important role in UCMF: increased urban temperatures, decreased windspeeds and

570 increased vertical turbulent diffusion. While elevated surface temperatures favor photochemistry, they also result in stronger dry-deposition as showed by Huszar et al. (2018a). Regarding the wind-speed and turbulence effect, they are counteracting which is seen in our results too especially for $SO_2$. For some of the cities, the impact is positive meaning that the reduction of winds results in the emitted material remaining close to the sources. This was previously seen also by Huszar et al. (2018b) where the turbulence and wind effects were strongly competing. Our results also showed that the trade-off between wind

and turbulence effects depends also on how the model simulates the UCMF components and in our results, WRF produced somewhat stronger increase in turbulence due to UCMF and weak wind reduction compared to RegCM. For ozone, the UCMF increased ozone by 2 ppbv, which is in line with previous finding in Huszar et al. (2018a), although they included also the effect of BVOC emissions modifications which was treated here separately (see further). Due to urbanization, similar increase was obtained by Martilli et al. (2003); Jiang et al. (2008); Xie et al. (2016a) or Jacobson et al. (2015). Some authors found

somewhat larger increases for ozone (e.g. Ryu et al., 2013, for Seoul), but they adopted higher resolutions for the cities in focus and thus obtained higher peak impacts in urban centres (as seen in e.g. Huszar et al., 2020b, too).

The diurnal cycle for the "DMET" impact shows a very characteristic pattern. In case of primary pollutants ($NO_2$ and $SO_2$) the decrease is strongest during evening hours. This can be explained by the largest absolute values during evening hours which is further closely related to the maximum impact of emissions – a similar finding was found by Huszar et al. (2018a)

585 and also by Huszar et al. (2018b) for primary aerosol components. Indeed, the amount of gases transported due to enhanced turbulence is proportional to the absolute concentrations and these are highest during evening hours due to strong emissions during transport rush hours. For ozone, the diurnal pattern contains a maximum during evening hours corresponding to largest impact on $NO_2$. This justifies the argument that the UCMF induced ozone increase is mainly caused by reduced NOx due to strong urban dilution and consequent reduced titration.

Besides the strong and well documented air-quality effects of urban emissions ("DEMIS") and UCMF ("DMET"), our study also looked at two other contributors to RUT, which were expected to be smaller but which were not yet quantified in detail.

Our study, at least by the knowledge of the authors, is among the firsts that explicitly investigated the effect of urbanization from the perspective of change in dry-deposition ("DLU_D") and we also looked at the effect of the urbanization induced changes in BVOC emissions, which was examined only partly in previous studies (e.g. Huszar et al., 2018a; Li et al., 2019).

The impact due to modified dry-deposition shows for $NO_2$ a distinct picture between summer and winter. For both seasons, reduced deposition velocities were modelled with stronger decreases in summer (the WRF driven CAMx results are not shown as they differ from the RegCM driven only slightly). Reduced deposition velocities result in higher concentrations which is opposite to what was modelled. To better understand what controls the $NO_2$ budget, we have to consider the simultaneous effect of ozone changes due to changes in dry-deposition. Our results showed strong increases in ozone concentrations which is

probably caused by suppressed dry-deposition (for winter too, not shown in this manuscript). This is expected as many studies showed strong dependence of both ozone concentrations on ozone deposition (Tao et al., 2013; Park et al., 2014) and ozone deposition on the landuse information (Mcdonald-Buller et al., 2001). When examining the concentration response to changed dry-deposition, one has to consider the indirect impact due to other influenced pollutants and probably pollutants responsible for $NO_2$ removal (e.g. by reaction $NO_2 + OH$ forming nitric acid or by $NO_2 + O_3$ forming nitrate radical) were impacted

by weaker dry-deposition (as seen for ozone) resulting in decreases in $NO_2$, outweighing the direct impact of dry-deposition change, as seen for winter. Another factor playing role in decreases of $NO_2$ can be in the much larger (by 50%) deposition velocities for nitric acid ($HNO_3$) in the Zhang model for urban landuse type compared to crops or similar rural landuse (i.e. "Nourban" case). Large dry-deposition for $HNO_3$ in turn results in decrease of this compound which reduces the recycling of $NO_2$ from it (by photolysis). On the other hand in summer, such effects can amplify the impact. In this season the deposition

induced ozone changes played probably a role in the $NO_2$ budget. It has to realized that a major pathway of $NO_2$ chemistry in cities is oxidation of NO with ozone ($NO + O_3 \rightarrow NO_2$). Increased ozone concentrations thus results in more NO oxidizing to $NO_2$.

    In case of sulfur dioxide, deposition increased in winter which resulted in clear decrease of near-surface concentration. The dry-deposition of sulfur strongly differs between wet a dry-soils (Hardacre et al., 2021) and according to Zhang et al. (2003)

which provided the dry-deposition scheme we adopted, the deposition velocities (DV) are higher for urban areas than for crops or similar rural landuse type (which was considered in the "Nourban" case). In winter, soils are very often wet which could result in increase in DVs (especially during night as seen in our results) and the consequent decrease of concentrations. During summer, DVs decreased for $SO_2$, however, there is no clear increase of concentrations, i.e. almost no change in RegCM driven simulations and even some decrease in the WRF driven ones. This can be explained again by the impact of deposition on other

chemical species which cause removal of $SO_2$, typically the oxidation by OH radical.

    The impact of BVOC emission changes ("DBVOC") is straightforward and expected for ozone, i.e. a decrease by 0.5-1 ppbv. BVOC emissions decreased due to urbanization related reduction of vegetation (i.e. reduction of vegetation fraction and leaf-area-index) and increased due to higher urban temperatures (within the action of the UCMF). This latter effect was smaller resulting in the dominance of the first effect and an overall decrease of emissions, a similar result as in e.g. Li et al.

(2019). As ozone chemistry in cities in Europe (and also North American and Asian megacities) is characterized by VOC-controlled regime with high NOx/VOC ratio (Beekmann and Vautard, 2010; Xue et al., 2014), ozone quickly responds to

changes in VOC emissions, i.e. it decreases with decreasing BVOC emissions. This is in accordance with previous studies: e.g. Song et al. (2008) reported an 10% decrease in ozone concentrations. The reduction in ozone was shown to be largest during daytime, which is in accordance with the largest BVOC emissions. Previously, Huszar et al. (2018a) reported ozone increases due to BVOC changes due to UCMF alone (i.e. not considering the impact of reduced vegetation) of order of up to 0.1 ppbv. Our study showed that if vegetation modifications related to urbanization are added, this increase is out-weighted by a much stronger decrease due to lower BVOC emissions.

Simultaneously with the "DBVOC" related ozone decrease, we calculated a small summer increase of $NO_2$ by about 0.01 ppbv. This cannot be explained by a potentially reduced NO concentrations and suppressed $NO_2$ formation with the reaction of ozone (titration) as NO also increased slightly as the result of "DBVOC" (not shown explicitly in this study) and moreover, soil NOx emissions in MEGAN also decreased slightly due to urban landuse transformation. Reduced BVOC emissions result in reduced peroxy-radical ($RO_2$) concentrations, which is an important oxidation pathway to form $NO_2$ from NO (Geng et al., 2011) and would result in decrease of $NO_2$. There must therefor exist another compensating mechanism responsible for NOx increase and this is probably the reduced concentrations of NOx sinks. One of the important urban contributors to this are the PANs (peroxy-acetyl nitrates) and as biogenic VOCs are a major contributor to urban PAN concentrations, it can be expected that with decreased BVOC emissions, the PAN sink is reduced resulting in higher NOx concentrations (Fischer et al., 2014; Toma et al., 2019). Another reasons can lie in the reaction with OH radical, which is reduced if ozone is reduced. In short, the relatively small positive $NO_2$ response to urbanization induced biogenic emissions changes is a probably a simultaneous acting of multiple indirect chemical pathways and deeper process based analysis should be performed to explicitly show the contribution and trade-off of each of them. The strongest changes modelled for Milan, Italy can be explained by its relatively warm climate and large size making the BVOC emission reduction strong and thus having a stronger effect on ozone and $NO_2$ (via secondary effects discussed above).

The diurnal patterns of different RUT contributors are explainable by the typical diurnal patterns of anthropogenic emissions and also by the UCMF's diurnal cycle. For the impact of urban emissions only, the "DEMIS" impact clearly resembles the emissions which peak in cities at morning and evening rush hours (Huszar et al., 2021). The impact of urbanization induced meteorological changes (UCMF) are governed by increased vertical eddy-diffusion which is strongest during daytime (Huszar et al., 2020a) and causes decrease of primary pollutant concentrations and increase of ozone (reduced titration; (Huszar et al., 2018a)). As the impact is proportional to the absolute concentrations, it is largest again during morning and (especially) early evening rush hours, during which usually even stronger vertical mixing occurs. A somewhat more complex explanation is needed regarding the diurnal pattern of the landuse change and dry-deposition impact. For ozone, a strong decrease occurs during day-time which is related to high day-time absolute values and stronger decrease of deposition velocities during the day due to urbanized landsurface. Over vegetated surface, ozone dry-deposition is determined mainly by stomatal resistance which is lowest during daytime meaning that the ozone dry deposition velocities are highest during the day (Park et al., 2014). When the surface is urbanized, the vegetation's role in dry deposition becomes very small and this daytime peak of DV disappears which results in the strongest decrease in DV during the day. The diurnal cycle of the "DLU_D" for $NO_2$ is probably the combined effect of decreased DVs which are strongest during the day and logically lead to increased concentrations, and

the reduction due to dry-deposition induced increase of ozone concentrations (see above how this might influence the $NO_2$ concentrations). As this latter occurs during noon-time, the resulting shape of diurnal pattern of "DLU_D" for nitrogen dioxide has a double peak. Finally, the diurnal cycle of the response of $SO_2$ to the dry-deposition changes due to urban land-surface has a clear daytime peak during winter when the impact is smallest. This is in line with the smallest increase of DV for $SO_2$ which occurs during daytime. During summer, the shape of the diurnal cycle for the impact on $SO_2$ is similar, however the values are higher making the peaks reaching positive values. These can be attributed the summer decrease of DV peaking during daytime, which leads to increase of concentrations. A more interesting feature is that the impact goes to negative values during nighttime for this pollutant. This cannot be explained by the DVs alone and, as already said above, some secondary chemical effect play role, e.g. increased ozone concentration lead to increased OH radical which can lead to increased oxidation of $SO_2$, as already suggested earlier. The diurnal impact of "DBVOC" for ozone in summer has a clear daytime negative peak, which is connected to the highest BVOC emissions occurring this time of the day. As the most important factor in the urbanization impact on BVOC is the reduction of vegetation, the highest reduction of emissions is expected to occur during daytime and this results, in a VOC-limited environment, in a stronger suppression of ozone production (i.e. leading to reduction). In case of $NO_2$, the "DBVOC" impact probably connected to decreased PAN which is an important sink for NOx (as already said above). As the absolute NOx concentrations peak during evening rush hours, this sink has the strongest effect during this time creating the evening peak of "DBVOC" for $NO_2$.

Our results further showed that the magnitude of each contributor for Prague is higher at the 1 km resolution nested domain than at 9 km resolution. This was already seen in previous studies focusing on the impact of UCMF in extreme air pollution (Huszar et al., 2020b) and concerned also the magnitude of the UCMF itself (e.g. the temperature increases due to urbanization have a higher city core peak at higher resolution). This was further seen regarding the impact of urban emissions only in Huszar et al. (2021). It can be assumed that if a nested domain approach at high resolution was applied to other cities, the impacts would have a stronger city core peak than at 9 km resolution.

To summarize our finding, we showed on an ensemble of 19 Central European cities that the strongest contributors to the impact of rural-to-urban transformation are the urban emissions themselves (increase concentrations for nitrogen dioxide and sulfur dioxide and decrease for ozone) and the urban canopy meteorological forcing (decreases the concentration of primary pollutants and increases those of ozone). They have to be accounted for simultaneously as the impact of urban emissions without considering UCMF can lead to overestimation of the impact (Huszar et al., 2021). Additionally, we quantified two weaker contributors. The effect of modified landuse on dry-deposition and the effect of modified biogenic emissions have one order weaker magnitude than emissions and the UCMF. However, we showed that for summer ozone, these are strong and of comparable order than the two strongest impacts. In other words, when analyzing the overall impact of urbanization on air-pollution for ozone, all four contributors have to be accounted for having a similar order of magnitude while for primary gas-phase pollutants (i.e. $NO_2$ and $SO_2$), the two weaker contributors are by at least one order of magnitude smaller and the error made is small if they are neglected.

Finally, we must stress that we focused on cities from relatively small region meaning the cities does not constitute substantially different climate and the typical "rural" vegetation was considered as crop. In other parts of the world, e.g. tropical areas

the rural-to-urban transformation takes place over diffent vegetation cover and e.g. the impact of modified BVOC emissions could be much stronger than in the case of central European cities. Further, it has to be noted that some secondary effects of modified pollutant concentrations can potentially play also a role via the direct and indirect radiative effect of emissions. The direct effect of aerosols can alter photolysis rates and temperatures influencing air chemistry (Han et al., 2020; Wang et al., 2022). It was further shown by many that aerosol emitted by urban areas modulates the vertical structure of the atmosphere resulting in modification of stability and/or convection (Miao et al., 2020; Slater et al., 2022; Fan et al., 2020; Yu et al., 2020) which in turn can modify the vertical mixing or the precipitation (Zhou et al., 2020; López-Romero et al., 2021), which finally feedbacks to influence on species concentration via wet-deposition and mixing. Our study was an offline coupled one meaning that no feedbacks from species concentrations via radiation and cloud/rain microphysics were accounted for. These studies however indicate, that to obtain an even more comprehensive picture of the total RUT impact, these secondary effects has to be taken into account too in the future.

*Code and data availability.* The RegCM4.7 model is freely available for public use at https://github.com/ICTP/RegCM (last access Aug 17, 2022) (Giuliani, 2021). CAMx version 7.10 is available at http://www.camx.com/download/default.aspx (last access Aug 17, 2022) (ENVIRON, 2020).The RegCM2CAMx meteorological preprocessor used to convert RegCM outputs to CAMx inputs and the MEGAN v2.10 code as used by the authors is available upon request from the main author. The complete model configuration and all the simulated data (3-dimensional hourly data) used for the analysis are stored at the Dept. of Atmospheric Physics of the Charles University data storage facilities (about 5TB) and are available upon request from the main author.

*Author contributions.* PH created the concept and designed the experiments, PH and JK performed the model simulations, LB , ML and KS contributed to input data preparation, model configuration and analysis of the outputs, APPP contributed to the validation, ALL authors contributed to the manuscript

*Competing interests.* The authors declare that they have no conflict of interest.

*Acknowledgements.* This work has been funded by the Czech Science Foundation (GACR) project No. 19-10747Y and partly by projects PROGRES Q47 and SVV 260581/2022 – Programmes of Charles University. We further acknowledge the CAMS-REG-APv1.1 emissions dataset provided by the Copernicus Atmosphere Monitoring Service, the Air Pollution Sources Register (REZZO) dataset provided by the Czech Hydrometeorological Institute and the ATEM Traffic Emissions dataset provided by ATEM (Studio of ecological models). We also acknowledge the providers of AirBase European Air Quality data (http://www.eea.europa.eu/data-and-maps/data/aqereporting-1).

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

**Table 1.** The list of RCM simulations performed.

| Regional Climate Model (RCM) simulations | | |
|---|---|---|
| Model | Urbanization[a] | Resolution[km] |
| RegCM | Urban | 9/3/1[b] |
| RegCM | Nourban | 9/3/1 |
| WRF | Urban | 9 |
| WRF | Nourban | 9 |

[a]Information whether urban land-surface was considered. "Nourban" means replacing the urban landsurface by crops
[b]Simulation performed in a nested way at 9, 3 and 1 km horizontal grid resolution.

**Table 2.** The list of CTM simulations performed with the information of the effects considered. The "Driving meteorology" and "Driving meteorology (BVOC)" columns correspond to the information from Tab. 1 above, i.e. which RCM simulation from the "Urban"/"Nourban" pair was used.

| | Regional Chemistry Transport Model (CTM) simulations | | | | | |
|---|---|---|---|---|---|---|
| | Experiment | Driving meteorology | Urban emissions | Landuse (deposition) | Landuse (BVOC) | Driving meteorology (BVOC) |
| 1 | ENNrrN (Reference) | Nourban | No | Rural | Rural | Nourban[a] |
| 2 | ENYrrN | Nourban | Yes | Rural | Rural | Nourban |
| 3 | ENYurN | Nourban | Yes | Urban | Rural | Nourban |
| 4 | ENYuuN | Nourban | Yes | Urban | Urban | Nourban |
| 5 | ENYuuU | Nourban | Yes | Urban | Urban | Urban |
| 6 | EUYuuU | Urban | Yes | Urban | Urban | Urban |

[a]information whether the meteorology driving the MEGAN model accounted for the UCMF

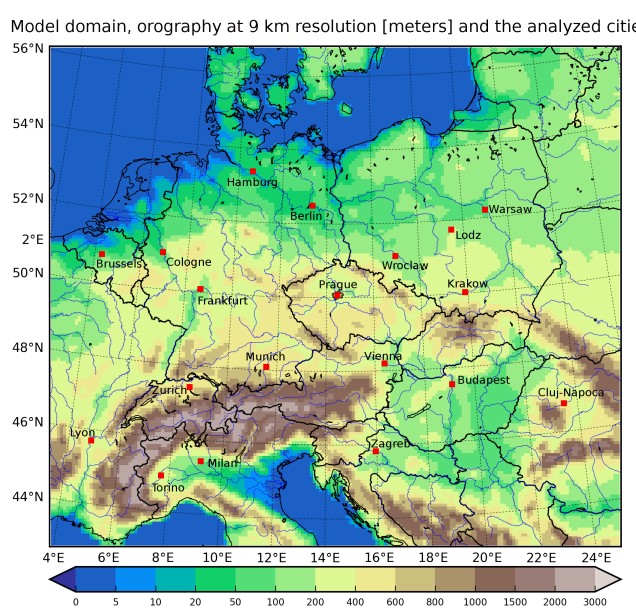

**Figure 1.** The 9 km x 9 km resolution model domain and the resolved terrain in meters including the cities analyzed in the study (red squares).

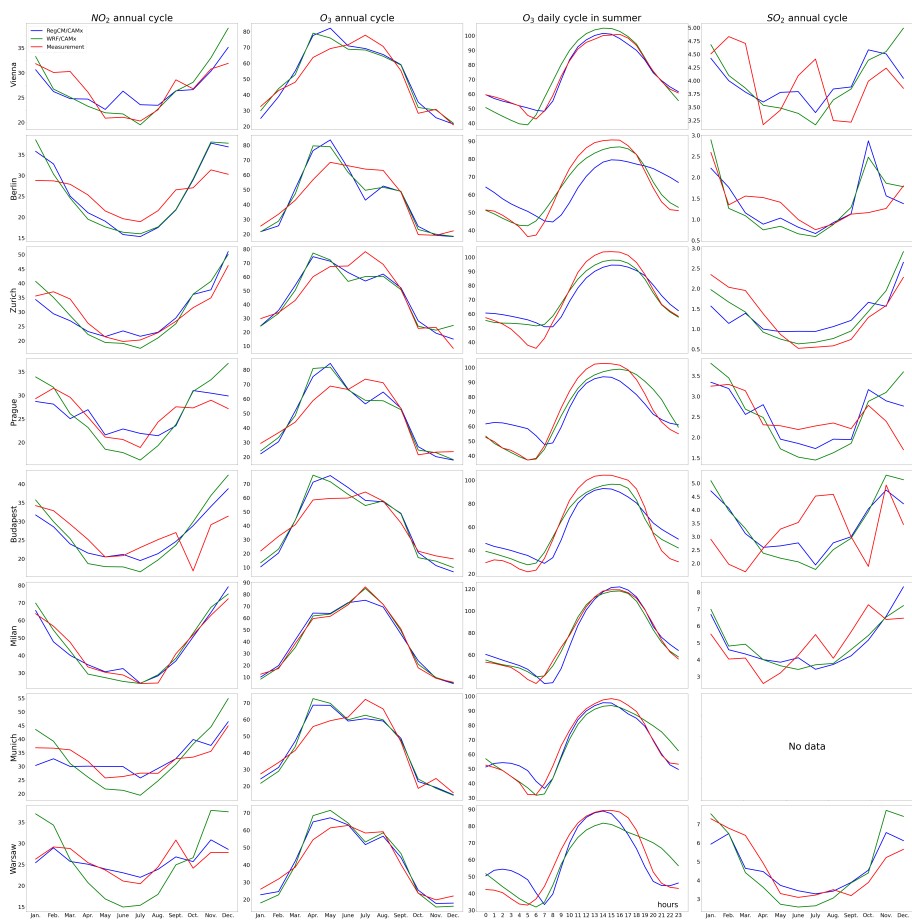

**Figure 2.** Comparison of modelled (blue – RegCM/CAMx; green – WRF/CAMx) and measured (red; AirBase data) urban and suburban average monthly concentrations of $NO_2$ (1st column) and $O_3$ (2nd column), the average JJA diurnal cycle of $O_3$ (3rd column) and the average monthly concentrations of $SO_2$ (4th column) for eight different cities selected from the total 19 considered in the study, namely Berlin, Budapest, Milan, Munich, Prague, Zurich, Vienna and Warsaw. Units in $\mu gm^{-3}$. Data are averaged across all available urban and suburban type background stations within the chosen city. No data for $SO_2$ in Munich as no corresponding measuring station was available.

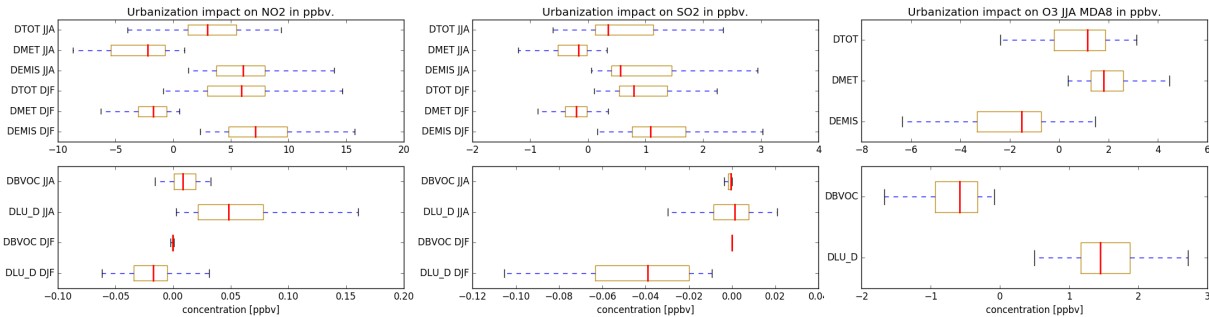

**Figure 3.** The 2015-2016 DJF and JJA averaged impact of each contributor to the rural-urban transformation including the total impact averaged over all chosen city for $NO_2$, $SO_2$ and $O_3$ in ppbv. In case of ozone, only the summer averaged MDA8 (maximum daily 8-hour average) is shown. The boxplots show the 25% to 75% quantiles including the minimum and maximum value across all cities. The red line shows the median. Values are taken from model grid-cell that covers the city center. The upper sub-figures show the two main contributors including the total impact ("DEMIS", "DMET" and "DTOT") while the lower one the minor contributors ("DLU_D" and "DBVOC")

.

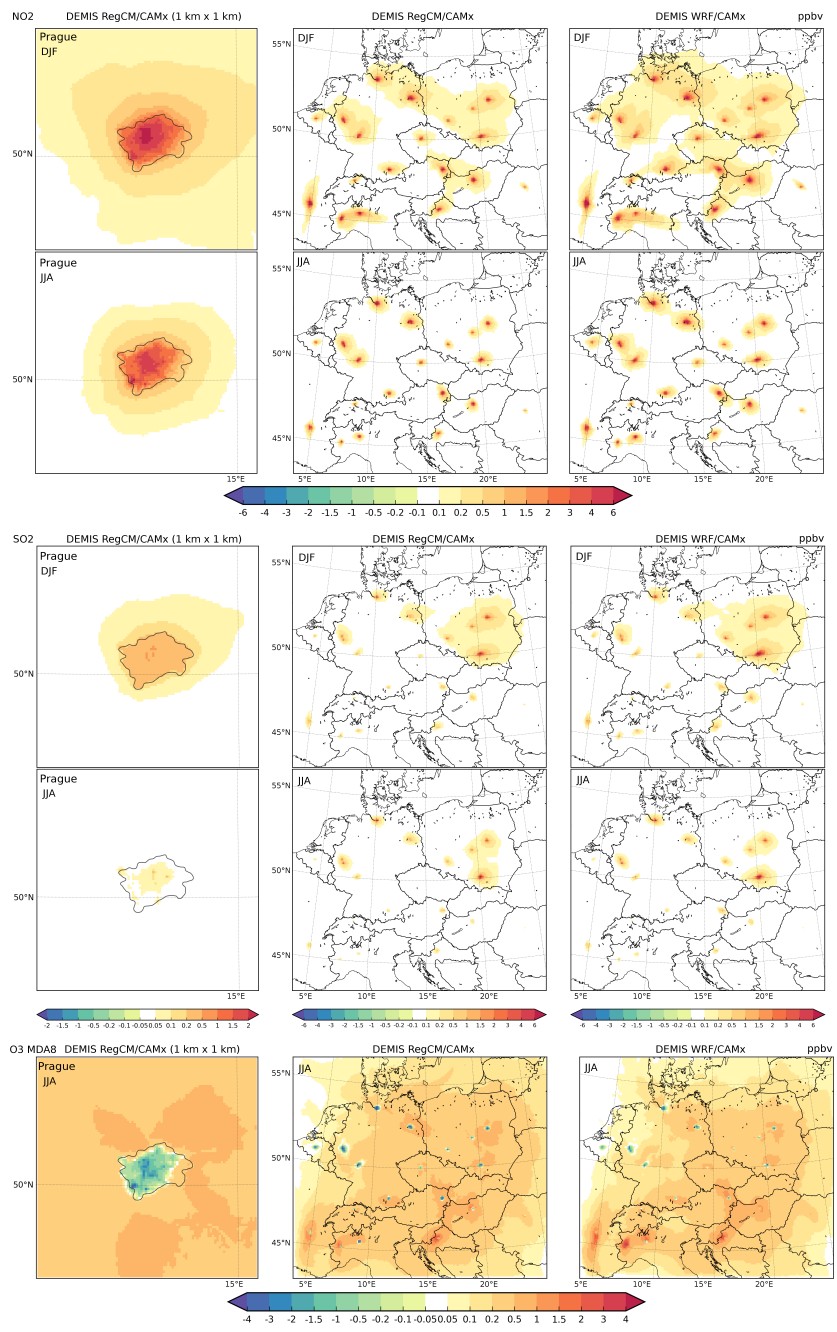

**Figure 4.** The spatial distribution of the 2015-2016 average emission impact "DEMIS" for $NO_2$ DJF and JJA (1st and 2nd row), $SO_2$ DJF and JJA (3rd and 4th row) and JJA MDA8 $O_3$ (5th row). Columns represent the results from the 1 km RegCM/CAMx (detail of Prague), the 9 km RegCM/CAMx and the 9 km WRF/CAMx simulations. Units in ppbv.

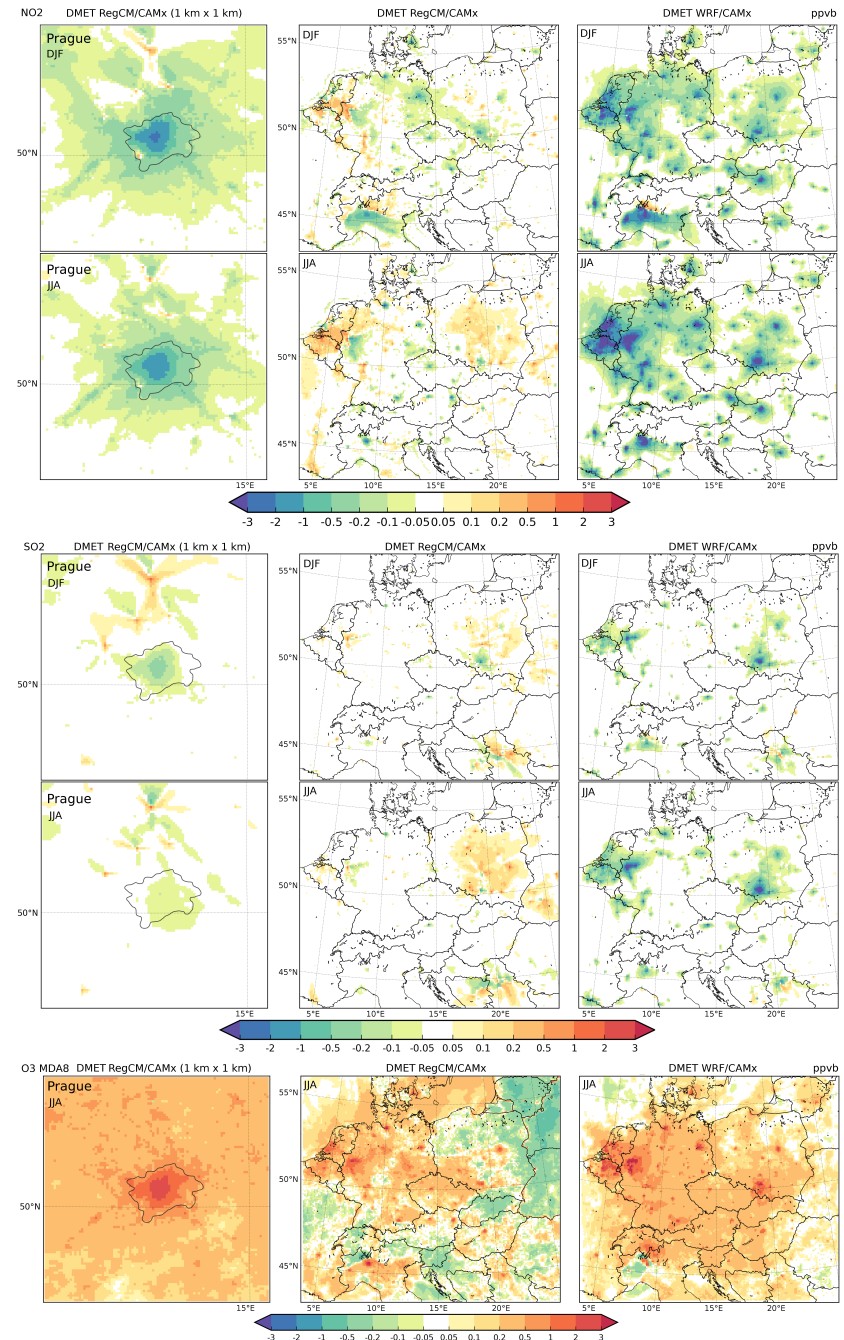

**Figure 5.** The spatial distribution of the 2015-2016 average impact of the urban canopy meteorological forcing (UCMF) "DMET" on $NO_2$ DJF and JJA (1st and 2nd row), $SO_2$ DJF and JJA (3rd and 4th row) and JJA MDA8 $O_3$ (5th row). Columns represent the results from the 1 km RegCM/CAMx (detail of Prague), the 9 km RegCM/CAMx and the 9 km WRF/CAMx simulations. Units in ppbv.

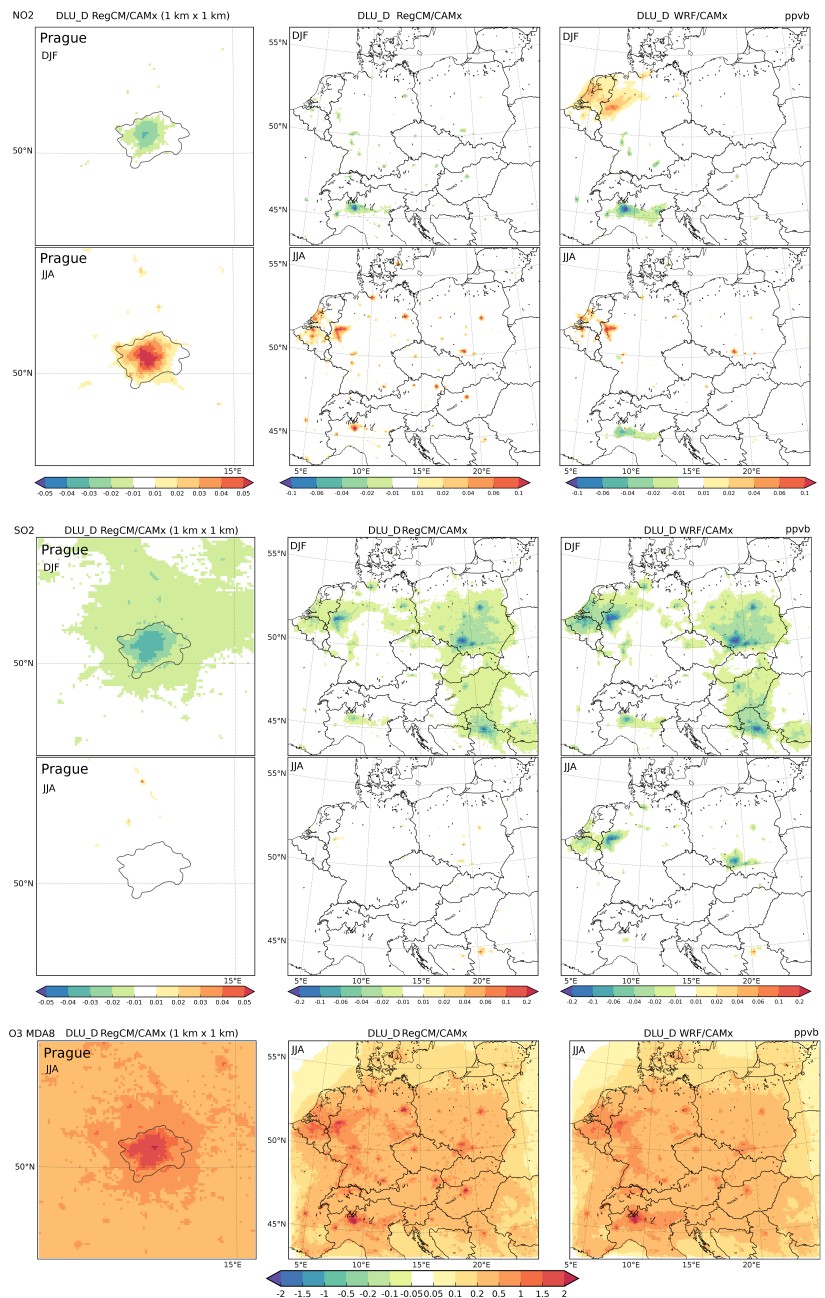

**Figure 6.** The spatial distribution of the 2015-2016 average impact of the urban land-cover via dry-deposition modifications ("DLU_D") on NO$_2$ DJF and JJA (1st and 2nd row), SO$_2$ DJF and JJA (3rd and 4th row) and JJA MDA8 O$_3$ (5th row). Columns represent the results from the 1 km RegCM/CAMx (detail of Prague), the 9 km RegCM/CAMx and the 9 km WRF/CAMx simulations. Units in ppbv.

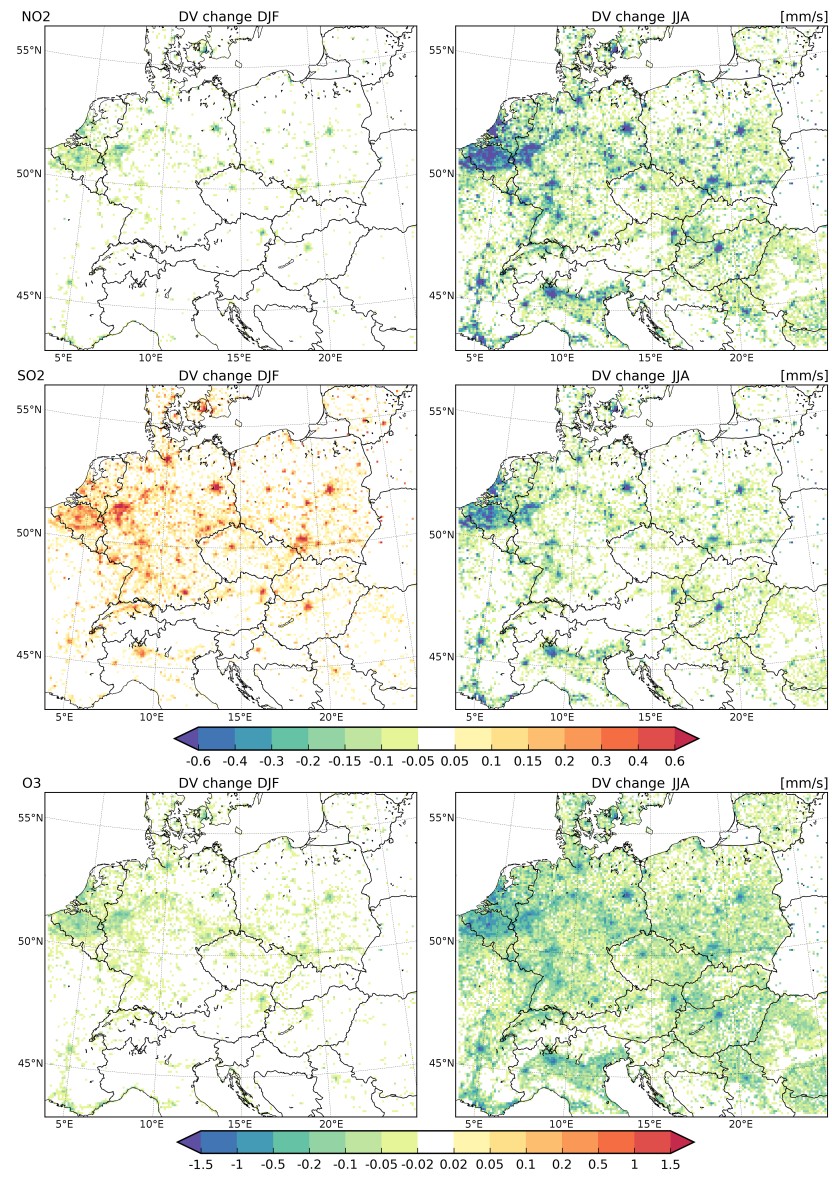

**Figure 7.** The spatial distribution of the 2015-2016 average impact of the urban land-cover on deposition velocities of $NO_2$ (1st row), $SO_2$ (2nd row) and $O_3$ (3rd row) for DJF (left) and JJA (right) in $mm.s^{-1}$ for the RegCM driven 9 km CAMx simulations.

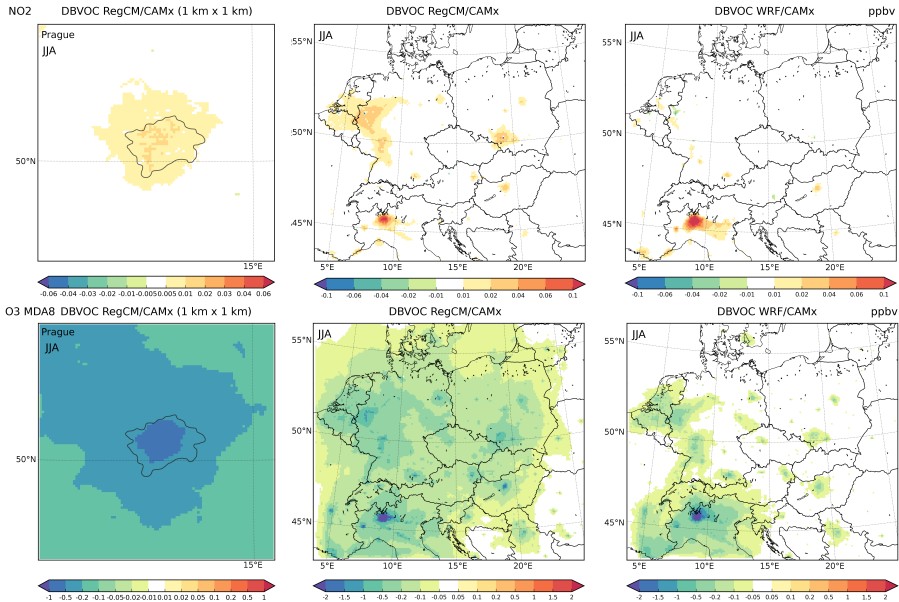

**Figure 8.** The spatial distribution of the 2015-2016 average impact of urbanization due to modifications of biogenic emissions ("DBVOC") on JJA average $NO_2$ (upper row) and summer MDA8 $O_3$ (lower row). Columns represent the results from the 1 km RegCM/CAMx (detail of Prague), the 9 km RegCM/CAMx and the 9 km WRF/CAMx simulations. Units in ppbv.

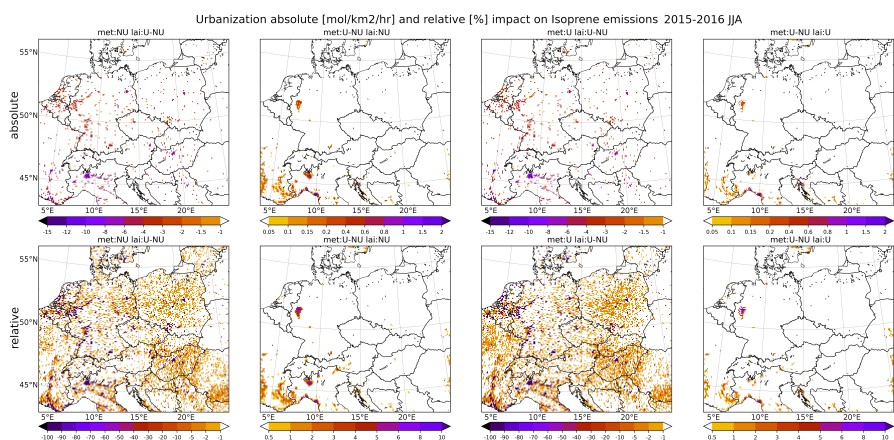

**Figure 9.** The absolute (upper row; units $mol.km^{-2}.hr^{-1}$) and relative change (lower row; units in %) of 2015-2016 JJA averaged isoprene (ISOP) emissions decomposed into the part caused by reduced vegetation (via leaf-area-index; "DBVOC_L") and the part caused by modified meteorology ("DBVOC_M"). The 1st and 3rd columns show the change due to "DBVOC_L" taking the rural (NU) and urban (U) meteorological conditions as a reference, respectively. In the 2nd and 4th columns, the changes due to urban meteorological effects (UCMF) are shown ("DBVOC_M") taking the rural (NU) and urban (U) vegetation as a reference, respectively.

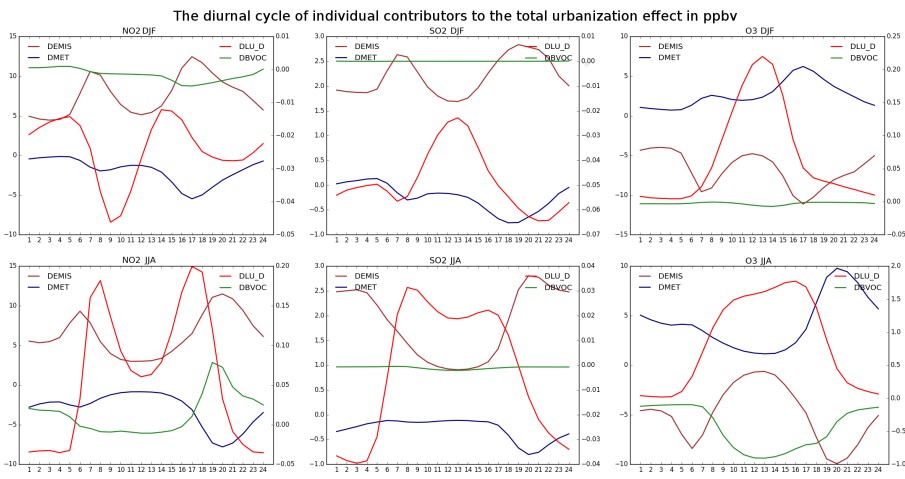

**Figure 10.** The 2015-2016 average diurnal cycle of the individual contributors to RUT for $NO_2$ (left), $SO_2$ (middle) and $O_3$ (right) as DJF (upper row) and JJA (bottom row) average. The brown and blue lines stand for the two stronger contributors ("DEMIS" and "DMET", left y-axis), while red and green stand for the weaker contributors "DLU_D" and "DBVOC" (right y-axis). Units in ppbv. Times in UTC (i.e. the local time is +2 hours in JJA and +1 hours in DJF).

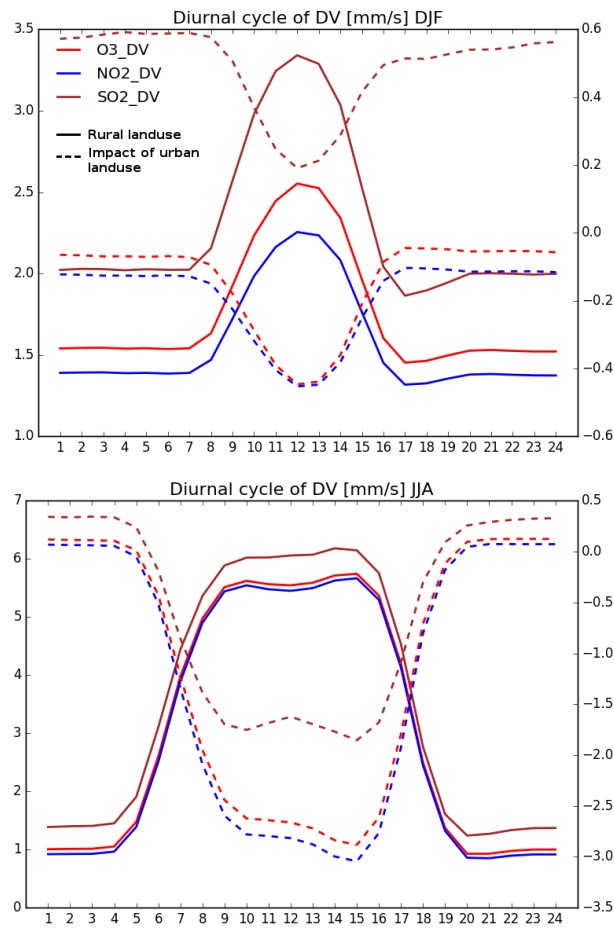

**Figure 11.** The diurnal cycle of the rural deposition velocities (DV; solid lines; left y-axis) and the impact of urbanization (dashed lines; right y-axis) for 2015-2016 DJF (up) and JJA (bottom) for $NO_2$ (blue), $SO_2$ (brown) and $O_3$ (red) in $mm.s^{-1}$.