# Peer review of "Impact of urbanization on gas-phase pollutant concentrations: a regional scale, model based analysis of the contributing factors"

_Atmospheric Chemistry and Physics, 2022_

## Author Response (AR1)

**Authors responses on anonymous Referees reports on "Impact of urbanization on gas-phase pollutant concentrations: a regional scale, model based analysis of the contributing factors"**

by Huszar et al. (acp-2022-337)

We thank both anonymous referees for their detailed review and all the comments. We will address each of them and our point-by-point responses follow below. Reviewer's comments are italicized.

**Referee #1's comments:**

*The authors investigate the impacts of some key assumptions about representation of urban anthropogenic emissions, dry dep and biogenic emissions, and meteorology from the urban canopy on the fine scale representation of ozone, SO2, and NO2 across a part of Europe with a lot of cities using an uncoupled modeling frameworks employing offline meteorology from two different regional climate models. They look at winter and summer averages for SO2 and NO2, and summer MDA8 ozone.*

*This is a useful contribution to the peer reviewed literature, but the framing of the paper and its implications need work to get the paper to be publication ready.*

*Major issues.*

*The authors sometimes overly generalize (& in some cases w/ excessively strong statements).*

*For example,*

> *•I find that the use of the term 'the urban canopy meteorological forcing' is too general. It's hard to imagine how the authors can say that the impact of the urban canopy meteorological forcing' always decreases primary pollutants but increases ozone. to me, this is equivalent to saying: 'meteorology always increases ozone'.*
> *•The statements "these are the two major drivers of urban air pollution", "the two minor contributors can be neglected" & "it is clear that the main driver affecting…" are quite strong... I don't believe they should be this strong, based on the evidence (or lack thereof) presented by this work. The authors are testing very specific assumptions about different processes.*
> *•The discussion of local versus rural/other cities' influences on air pollution in the Intro seems limited. some pollutants' distributions and production are very regional scale, so separating sources into local vs. rural doesn't make much sense. Additionally, it seems strong to say that 'air pollution in cities is mainly determined by local sources'… for ozone, background levels in cities should be an important fraction of the total ozone.*

Authors response: The term „urban canopy meteorological forcing" (UCMF) was introduced and defined in Huszar et al. (2020) as the forcing that the urban canopy has on the city scale meteorological conditions and thus on air-chemistry via modified transport, increased turbulence, increased temperatures etc. In other words, it is the urbanization induced changes of meteorological conditions. If we thus refer to the impact of UCMF, we do not refer to the impact of urban meteorology as it is, but the impact of the urbanization induced changes only. In Huszar et al. (2018a,b) and in many other cited literature it is shown that one of the most important components of UCMF is the increased turbulence which causes decrease of primary pollutants over-weighting the decreased wind speeds and reduced dilution (as another component of UCMF). On the other hand, for ozone, which is a secondary pollutant, the effect of UCMF is more complicated as UCMF acts on both ozone itself and its precursors. The net result in this case is an increase with strong contribution of decreases NOx concentrations and reduced titration (see Huszar et al., 2018a). Consequently, we can indeed say that UCMF increases ozone. We however admit that we cannot write „secondary pollutant" in general, as this impact can be different for

different secondary pollutants (e.g. Huszar et al.,2018b showed decreases of secondary inorganic aerosol due to UCMF), so we modified the text to be more specific.

Regarding the strong statement of „These are the two major drivers of urban air pollution"..  we refer here to the fact the if we consider the total impact of urbanization (i.e. the rural to urban transformation) on air pollution, we can clearly state the the two most important factor that constitutes this impact are the urban emissions themselves and the UCMF. We are not stating here that e.g. the background state or rural emissions do not play role in urban air pollution. We are only interested in the role of urbanization (rural emissions and the background air pollution is the reference base state), i.e. if a rural area is urbanized, its air pollution is significantly modified and the most important factors in this modification are the emissions and UCMF. In the revised manuscript, we rephrased some of the above criticized statements to be more precise.

Regarding the statements about the two minors contributors, we admit that although their contribution is much smaller, based on this work we cannot state that they can be neglected. This is indeed a strong statement and we removed it from the text (and rephrased the remaining text slightly).

As for the „*The discussion of local versus rural/other cities' influences on air pollution*", we have to stress, that the purpose of the analysis was to evaluate how urbanization affects the local air pollution, i.e. air pollution over the area where the urbanization occurs. The study is neither focusing on i) how urbanization  affects the regional scale air pollution (although some results are obtained in this regard too) nor on ii) how the rural/nourban areas affect the urban air pollution. This was also specified more clearly in the manuscript.

*There are scattered introductions to urban air pollution and ozone chemistry throughout the manuscript. It would be more helpful if this was concentrated in the Intro.*

Authors response: Some basic information on urban gas-phase air pollution (the three analyzed pollutants) is already provided in the Intro, but we wanted to keep the Intro short focusing rather on collecting the current knowledge about how urbanization affects the air-quality and what potential contributors it has. We admit, that some information regarding air pollution and ozone chemistry appears also elsewhere in the manuscript, but it was necessary in the context of that text, especially in the Discussion, where many of the modelled features and their interpretation required it.

*With respect to the introduction of the simulations in the Intro, it is not clear what the authors are investigating here. Are the authors using a global model? What is the resolution of the model? What is the base simulation? Are the authors assuming a preindustrial like state? Because this is a model study, it seems like this all should be very clear in the Intro.*

Authors response: We admit that we could do more in the Intro to make clear, what we are investigating, what is the reference state etc. We added some text to clarify this. E.g. we stressed that the purpose of the study is to evaluate the i) total impact of urbanization (``DTOT") and more importantly, the contribution of ii) each of the urbanization related impacts (i.e. ``DEMIS", ``DMET", ``DLU_D" and ``DBVOC")  over regional scale domain on the present day final urban air pollution levels using coupled regional climate and chemistry transport models applied at moderate horizontal resolution (9 km), so we stressed the facts that 1) regional scale models are used, for 2) present day conditions (including emissions and landuse), 3) the resolution used. We also added a sentence explicitly saying to which reference state we consider the effect of urbanization, i.e. the state "without cities", which means the urban landuse replaced by rural one and urban emissions removed (the way this is done is described in the Methodology and within an answer to the referees comment further below).

*[Later in the methods, the authors say: "in the first experiment where urban emissions are disregarded, we removed urban emissions only for the 19 cities chosen for the analysis". I think my fundamental question is: how are the authors defining what emissions are "urban"? what exactly are they investigating with this first simulation? This should be clear in the Intro, as well as the methods.]*

Authors response: this is indeed a fundamental question and we detailed in the Methodology section but also put a note in the Introduction, that: the urban emissions are defined as those falling within the city administrative boundaries (based on the ISO 3166 standard defining *countries and their subdivisions*.). The emission model we used (FUME) allows to mask out emissions based on a given geometry (shape), i.e. we could remove the emissions from all gridcells which completely lie within the geometry, while from those gridcells, across which the city boundary goes, only that part is made zero, which falls within.

The first simulation with CAMx (driven either with RegCM or WRF) is meant to be the reference (base) state, while with each further simulation we added one effect. The first effect added are the urban emissions, so the first two experiments allows us to evaluate the effect of urban emissions only (while the 2$^{nd}$ and 3$^{rd}$ allows to evaluate the effect of landuse on dry-dep, and so on, as defined by Eq. 4 to 8).

*How is the land use input required for the regional climate models similar/different from the input to the dry dep and biogenic emissions schemes? Seems like the authors need to emphasize that in some of their sensitivity simulations, the deposition/biogenic emissions are forced by meteorology that is decoupled from the land use type impacting the deposition/biogenic emissions.*

Authors response: The data for landuse information is the CORINE CLC 2012 landcover data (https://land.copernicus.eu/pan-european/corine-land-cover) which was used within RegCM and WRF as well as for the dry-deposition scheme in CAMx. As CORINE does not contain information needed for MEGAN, namely the annual cycle of the leaf-area-index (LAI), the plant functional types (PFT) and the plant emission potentials, these information were derived independently based on the data provided in Sindelarova et al.(2014, 2022). The meteorology driving the MEGAN model is either the one considering NOURBAN (rural) landuse, i.e. without the UCMF effects, or, the URBAN one, i.e. which does considers URBAN landuse (with the applied urban canopy model), see Tab. 1. (two pairs of RCM runs – with RegCM and with WRF).  With respect to the effect of urbanization on biogenic emissions and the consequent effects on air-quality, three experiments were important: No. 3, 4 and 5 (Tab. 2.). In experiment 3, we considered rural land information for MEGAN and NOURBAN (rural) meteorology, so in this experiment, biogenic emissions represent the "nourbanized" situation (i.e. the base/background state). On the other hand, in experiment 5, biogenic emissions are driven with urban meteorology (i.e. which considers urban effects – UCMF) and considers urban land information (reduced-zero LAI in our case). This experiment thus represents the full urbanized case concerning the biogenic emissions. To examine the partial effects of the modified land information and the UCMF on biogenic emissions, we performed the experiment 4, where the land information for MEGAN was "urban" while the meteorological driving fields were those considering NOURBAN (rural) landuse. We tried to emphasis this more clearly in the revised manuscript.

*The approach used by the authors to separate the roles of urban emissions, meteorology, dry dep, and BVOCs (detailed at the end of page 8) is full of assumptions. The effects are not additive; the authors probably should not assume equation (2), although generally looking at the difference between simulations to gauge sensitivity is fine. The authors need to adjust the framing of the work on this front (e.g., "individual components" – I wouldn't characterize as individual components when the components are tightly coupled in real life, and should be in the model).*

Authors response: as we said, the purpose of the study was to quantify the individual contributors to the total impact of urbanization as well as the total impact itself (DTOT). The total impact for a species i (expressed as its average concentration $c_i$) is calculated as the "full" (real, i.e. urbanized) situation minus the background (base, rural, reference) situation denoted $c_{i,rural}$. What we call the the total impact of urbanization is the difference, $c_i - c_{i,rural} = \Delta c_{i,RUT}$.(RUT - rural to urban transformation). This is our equation (1). As we were interested in decomposing the $\Delta c_{i,RUT}$ into different parts, we performed additional simulations in which we added different effects one-after-other as defined in Tab 2. And the effects themselves can be calculated as written in Eq. 4 to 8.

Indeed, if we add these four equations (5-8), we will get the right hand side of Eq 2. and the EUYuuU(6) – ENNrrN(1) difference. The later is exactly the total impact of urbanization (or $\Delta c_{i,RUT}$). So this implies that Eq 2 holds. The same argument justifies Eq 3, which is a additional decomposition of one of the contributors. So, the way how the effects are calculated makes them mathematically perfectly additive, as they are not added individually (separately) to the base simulation, but added gradually (in a cascading way). In other words, we have 6 simulations, SIM1, SIM2, SIM3, SIM4, SIM5 and SIM6, and what we state by equation 2 is that the Eq. SIM6-SIM1 = (SIM2-SIM1)+(SIM6-SIM5)+(SIM3-SIM2)+(SIM5-SIM3) holds, which is trivial.

We tried to clarify this more clearly in the revised manuscript and instead of "components" we will call them "contributors to RUT".

*For ozone, all the cities have very similar observed annual cycles for urban areas, and the model captures them. I'm not sure this is the best way to evaluate ozone for the purposes of this paper. The annual cycle is likely largely driven by regional scale phenomena rather than urban processes. It would be good to evaluate whether the model captures urban-suburban-rural gradients in ozone.*

Authors response: We admit that monthly means of ozone are not very representative in terms of model performance and the model's capability to resolve the hour-to-hour variation is important for this pollutant. Therefor we added the average summer diurnal cycle of the modelled and measured values as a further column in Fig. 2.

*Also, I wouldn't call an uncoupled modeling framework a novel approach.*

Authors response: the "novelty" of our approach was in decomposing the total impact of urbanization into individual contributions and this required often to decouple different effects from meteorology/landuse. This was not possible to do with a online coupled model, so the fact that the models (including the biogenic emissions mode) were offline coupled to the chemistry transport model was an necessity. However, we removed the word "novel" in our Introduction to avoid doubts.

*Minor issues.*

*There is a fair amount of grammar issues that a closer read or English editing could help (I didn't list these).*

Authors response: we revised our manuscript in this regard and corrected it with the help of an English editor.

*Other minor comments.*

*Intro.*

   •*What do the authors mean an 'artificial' surface?*

Authors response: This means not natural surface, i.e. surface with artificial objects like buildings, roads, parking etc.

   •*Ref for 60% of the population will live in urban areas*

Authors response: Reference added (UN(2018b))

   •*The authors shouldn't start a sentence with 'E.g.'*

Authors response: Corrected.

*•Sure, plants are a large sink of many gases, but that doesn't mean other surfaces can't be either. What's known about dry deposition to urban surfaces? Do we as a community know what deposition velocities should be over urban surfaces?*

Authors response: indeed, deposition to urban type landsurface in regional models is a complicated issue as urbanized surfaces are comprised by very different materials and the vegetation fraction of urban areas is also highly variable. The used dry deposition model (Zhang, 2003) considers some average value but we admit that this might be burdened by a high uncertainty given the fact that substantial differences between cities exist. Nevertheless, in their model, urban surface has a clearly smaller capability to remove gaseous material from the air compared to purely natural surface. However, some secondary pollutants like $HNO_3$, $HNO_4$, $H_2SO_4$, $H_2O_2$, HONO or also $NH_3$, the removal in case of wet canopies is higher for urban areas than for rural ones (e.g. crops) due to their high solubility and reactivity with solid surfaces. So in general, urbanization modifies the removal depending the the pollutant itself and the state of the canopy.

*•Again, yes, plants are a large source of BVOCs, but that doesn't mean that the plants that we do have in the cities aren't very efficient producers of BVOCs, or there aren't emissions of BVOCs from consumer products. (In the past couple of years, there has been evidence of both. To be clear, I'm not asking the authors to investigate efficient emissions of BVOCs from urban vegetation or BVOCs from consumer products, rather contextualize and motivate their study accurately/well)*

Authors response: indeed, urban vegetation like any other vegetation is a great producer of BVOC but when the rural surface is turned into urban one (during the RUT) than this vegetation is reduced (will not be zero, only reduced) and consequently the emissions of biogenic gases are also reduced.

*•Use of 'background' as defined as 'without urbanization' is not great, given there is clear definition of 'background ozone' (https://doi.org/10.1525/elementa.309)*

Authors response: we agree that "background ozone" has a well defined terminology, however we had to define a background (base) state here to which the individual urban effects are gradually added, therefor we prefer to use "background" for the non-urbanized situation.

*Methods*

*•The authors are taking grid-cell average meteorology and feeding the CTM with this? Needs to be super clear.*

Authors response: the meteorology provided by the driving RCM (RegCM or WRF) is defined in points – in case of state variables (temperature, humidity, cloud/rain water) these are the grid-cell centers, in case of wind components and diffusion coefficients these are defined at grid-cell faces. The state variables represent grid-cell averages in CAMx, the same for the concentrations. This was clarified in the revised manuscript.

*•What is the point of discussing CLMU in CLM? Is CLM coupled to RegCM4.7? Also, what do the authors mean by 'the traditional urban geometry approach is implemented'?*

Authors response: we admit that in the current formulation, the reader could have the impression that CLM4.5 (in which CLMU is included) is a separate model independent from RegCM, which is of course not true. CLM4.5 is the surface exchange model implemented in RegCM4.7 (replacing the older BATS surface scheme). We reformulated the sentence to clarify this.

*•It seems like it should be emphasized that the authors are looking at not only two different regional climate models, but additionally two different urban canopy parameterizations.*

Authors response: yes, this is true and we stressed this in the "Model simulations" section.

*•Do S&P 1998 and Zhang 2003 give wet deposition parameterizations, or just dry dep? Also, it seems like the authors should elaborate on how urban dry dep is simulated in CAMx and its key sensitivities.*

Authors response:  S&P stands for the wet-deposition while Zhang 2003 refers to the dry-dep scheme. This was clarified in the revised manuscript. We also added some key information how the deposition velocities are calculated and what resistances are considered as well as some notes on how these are modified if urban landsurface is considered and on what meteorological parameters are they dependent.

*•How do the authors know that the impact of soil NOx is small?*

Authors response: In our experiments to soil-NOx emissions are about two orders of magnitude smaller compared to the BVOC emissions (we plotted these emissions but not presented in the manuscript). Therefore we expected their effect to be much smaller, as well as the effect of their urbanization induced modifications.

*•What does 'showed that their long-term effect is rather small' mean?*

Authors response: We meant that the long-term (10yr in the mentioned study) radiative effects of urban pollutant emissions (i.e. not GHG emissions) is very small and is comprised mainly by the cooling effect of aerosol, while the effect of ozone modifications due to urban emissions are almost negligible. This was clarified in the revised manuscript.

*•Clarify that the nested domains are only for one of the regional climate models*

Authors response:  this is later clarified in the Model simulation section.

*•What does 'they found a rather small impact' mean (end of first paragraph of 2.2)?*

Authors response:  we extracted some more specific, quantitative information from the mentioned studies, i.e. the that the sensitivity of ozone concentrations to model resolution is about 10% with higher model resolutions producing smaller concentrations.

*•What does 'brings some accounting for the uncertainty related to the urban land-cover representation' mean?*

Authors respose:  it means that due to the fact that the landuse is represented differently in WRF and RegCM, dominant landuse category vs. fractional, respectively, partly urbanized surfaces can be differently accounted for in the two models: for example. if 60% of  the gridcell is covered by urban surface than in WRF (dominant landuse), the total gridcell area is considered as "urban", while in RegCM only the 60% of the gridcell (fractional). This means, that some of the urban meteorological effects can be stronger (or different in more general fashion) in WRF.

*•How do they authors grid the irregularly shaped Czech emissions to the model? Is this where FUME comes in? If so, I'm confused why the descriptions are in separate paragraphs.*

Authors response: all emissions all interpolated by FUME, so the regular lat-lon CAMS data as well as the irregular Czech emissions. The reason is that the model grid is always different from the input data spatial distribution, regardless of the fact if these are regularly gridded or irregular shapes. In the revised manuscript, we reorganized the Paragraphs. Now all the information regarding anthropogenic emissions including FUME are in one paragraph.

•*I'm confused when the authors are discussing the "city" vs. "non-city" portions of the cross-boundary shapes. Is there a sub-grid distribution of emissions?*

Authors response: we meant that emission over emissions cells (regular in CAMS or irregular in the Czech emissions), which are divided by the boundary of the city, has to be split into the part that lies inside the city boundaries and the part lying outside. This is were the masking capability of FUME can be used.

•*In replacing the urban land with the rural crop land over the entire domain (not just the cities chosen), the authors may be changing background levels of air pollutants, not just local urban levels.*

Authors response: it is true that urban land use was replaced to rural one (crop) over the entire domain, but the effect on background levels is expected to be small for two reasons: 1) emissions from these areas were still considered, 2) the urban meteorological effects from these (minor) urban areas has rather small influence or air pollutants as the UCMF is also small (see e.g. Huszar et al.(2014) so the effects are important mainly around large urban centres we are interested in also in this manuscript).

*Results*

•*What's the first difference? I only see a reference to the second difference in the validation section. Also, if a different chemical mechanism is used, then that seems like a big change.*

Authors response: the first difference was the newer model versions. We reformulated this paragraph to more stress that there are indeed differences so some level of validation is necessary.

•*I don't know what the authors mean by "while all urban and suburban background stations were used from a subset of the analyzed cities". Why a subset? What does background mean here? Which sites were chosen for the evaluation & why? Needs to be clear.*

Authors response: we agree that optimal would have been to choose station for all urban areas. However we made this subselection in order to reduce the amount of subfigures. Further, instead of an average-over-all-cities, we rather wanted to show whether there are some notable differences between individual cities or the model biases/performance is similar over different city. Our selection of cities for validation (8 from the total of 19) followed the criteria being largest from the total of 19 cities chosen. Another criteria was to select cities which are more-or-less evenly distributed over the domain, so the validation conclusions from them can be applied to other cities. Lastly, we had to select cities with sufficient number of measuring stations (which correlates usually with the city size of course.) Regarding the selection of the station, we chose AirQualityStationArea = "urban" and "suburban", while only AirQualityStationType = "background" for the stations to fullfill. "Background" stations were needed for representativeness of the concentrations over a larger area (like gridcell).

•*It seems like there should be some text about the diel cycle scaling of the urban emissions in the methods.*

Authors response: we provide a reference for the temporal dissaggregation factors used to calculate hourly emissions from annual one (van der Gon et al.(2011)). These include a sector-specific diel cycle to obtain hourly

values from daily means (which are obtained using monthly factors and factors for week-days). As these are dependent only on the activity sector but not on the geographic location, they will be certainly not equally representative for all cities which could add some biases for some of them.

*Discussion and conclusion*

•*Word choice -- the authors selected four contributors based on previous work to investigate, they did not identify them based on their work …*

Authors response: we rephrased the sentence and removed the "identify".

•*A lot of the discussion, especially with respect to model evaluation, is framed as a comparison to the authors' previous work. This is not too helpful for the reader who is not familiar with that work. Can the authors focus on what knowledge is generated from this work first, and then compare (briefly) to previous work by the same author? In other words, the comparison should be secondary, not primary.*

Authors response:  we reorganized this paragraph to stress what "knowledge was generated" at first place and than how this knowledge relates to previous work or how it can be explained in the context of previous work. This was sometimes necessary as our experiments evolved from these and the updates in the experiments usually explained many of the model performance improvements.

•*I'm not sure how the authors can go from "in the case of SO2, the model is rather unable to correctly resolve the annual cycle of near-surface concentrations" to "in summary, we did not identify substantial model biases" within a couple of sentences…*

Authors response: we agree with the reviewer and removed this formulation.

•*I recommend cutting the "hints that the effect of urban emissions is well captured" part*

Authors response: removed.

•*Why is there stronger dry dep due to higher temperature? If there are meteorological sensitivities to the biogenic emissions and dry dep parameterizations, then they should be spelled out in the descriptions of the models.*

Authors response: Here we refer to Huszar et al. (2018a) who writes "In the dry-deposition scheme used (Zhang et al., 2003), the aerodynamic resistance ($r_a$) and the stomatal resistance ($r_s$) depend on temperature. The first is calculated according the near surface stability and mixing conditions following the Louis (1979) scheme incorporated within CAMx. Higher near surface temperatures result in reduced stability decreasing the corresponding resistance. The stomatal resistance decreases strongly with temperature as a result of more opened stomatas thus effective ozone uptake." To stress this temperature dependence, we added some sentences in the methods section from which implies that the temperature increase leads to increase in dry-deposition. The same was done for BVOC emissions, as these are also strongly dependent on near-surface temperatures.

•*If the piece about increasing in ozone due to suppressed dry dep is not shown in this manuscript, discussion is not merited in the intro. Too speculative.*

Authors response:Figure 7 clearly shows that ozone dry-dep is decreased due to urbanization related land-cover change so we can say the increase of ozone is probably due to this suppression.

*Table 1.*

  •*I think it would be clearer to say 'crop' rather than 'Nourban'*

Authors response: We prefer to use "Nourban" to be consistent with our previous studies and with the fact that it represent the land-surface without urbanization, i.e. rural land-surface. However, we added explicitly in the footnote that Nourban means crops.

*Table 2.*

  •*For parallel structure, do Landuse (deposition) vs. Landuse (BVOC) as the column names.*

Authors response: corrected. We also changed the last column name to Driving meteorology (BVOC)

  •*It's confusing to have 'nourban vs urban' for meteorology, and then 'rural vs. urban' for land use. Do the authors want to have different descriptors here? If so, please spell out why in the table footnotes and/or text.*

Authors response: Urban/Nourban for meteorology more relates to the effects, i.e. the differences in the meteorological parameters between the Urban and Nourban landcover.  In Tab 2 therefor we prefer to use Urban/Nourban to columns were we refer to the driving meteorology (to be consistent with Tab 1.). On the other hand, in the "Landuse" columns, we would like to stress that the landuse is "rural". So we prefer to use different descriptors.

*Figure 1.*

  •*Can this map include the model grid? (Also, are the two model grids the same?)*

Authors response: the map exactly corresponds to model grid (model domain) which is same for WRF/RegCM/CAMx. Adding the gridlines would make the figures very messy.

  •*Make city symbols larger.*

Authors response: Corrected.

  •*What does the [m] in the title mean?*

Authors response: Means "meters". Corrected.

  •*Why show terrain rather than land use type?*

Authors response: we agree that showing the urban landuse would be logical, however the main message from such a figure would be the location of the main urban centres, which is indicated in the actual figure too. Moreover, the urban landuse figure was already presented in Karlicky et al. 2020 (Fig 1 and 2) which uses the same domain and landuse data. So we rather decided to show a different information: the orography which is an important parameter modulated the regional climate.

*Figure 2*

   •*Are the observations from the same year as the model? Please specify*

Authors response: of course, observation are from the same date/time as the model data. Clarified in the manuscript.

   •*Can the authors put error bars on the observations to represent variability across stations examined?*

Authors response: as the WRF driven CAMx "lines" were added to the figures (see the response below), we think that the errorbars would make the (sub)figure(s) harder to read. Also, the data are from small amount of stations per city (usually 1 to 10 station per city) to make the errorbars representative. We therefore prefer the presentation without errorbars.

   •*Why is there only one model line on here? Shouldn't there be two (the CAMx simulations with different meteorology)?*

Authors response: the WRF driven CAMx simulations was added to the validation (green).

   •*I get that some authors like to convert the model values to whatever units the observations are in for model evaluation, but the inconsistency between the units of this plot and the ppbv used elsewhere makes contextualizing the model's limitations challenging.*

Authors response: it is true that the units for the results (ppbv) are different from the units in the validation (ug/m3) however we preferred to stay consistent with previous validation studies (ours and others too) where the model values were converted to the units of the measurements. Morever, in this conversion we had to use the modeled temperature and pressure values (needed for the ppbv ↔ ug/m3 conversion). However, we do not have the temperature/pressure data from the stations chosen, so we cannot correctly convert the measured data into ppbv, only vice-versa.

*Figure 3*

   •*The error bars show the maximum and minimum across grid cells, or across days?*

Authors response: they show the 25% and 75% percentiles as well as the minimum/maximum values across all the chosen cities. Clarified in the revised manuscript.

*Figures 4,5,6 and 8*

   •*Can the Prague panel be the first panel, and then the larger domain ones be the second and third panels? This would make it easier to compare the two larger domain panels...*

Authors response: We agree with the referee and reordered the columns so now the first one shows the detailed result for Prague. We adapted the captions and the text accordingly.

*Figure 10*

*•Are these changes, or absolute? I think the former because there are some negatives. These changes look huge, but I don't know what the baseline is. Maybe just show the individual simulations (rather than the changes)? This goes back to my saying that you can't really consider the differences in the simulations as additive.*

Authors response: these are the individual contributors to the total urbanization impact (defined by Eq.5-8), so we present differences of simulations (that's why we have negative values, indeed). There is no single baseline but the contributions are built on each other as mentioned earlier and defined by these equations. So by this definition these contributors are additive. The reason for not presenting the absolute values is that the difference between the simulations (defining these contributors) is often very small. This is the reason, why we have chosen a separate y-axis for the weaker contributors (DLU and DBVOC; the right axis).

*Figure 11*

*•It might be better to show both DV for urban vs. crop, because it's hard to contextualize the impact without knowing the absolute values (& many readers may be unfamiliar with the magnitude of DV.*

Authors response: we added the absolute DVs corresponding to rural landuse (crops) as solid lines (left y-axis) and kept the differences arising from urbanization  (i.e. the impact of urban landuse; dashed lines; right y-axis). Now the reader easily seas the difference and compare it to the absolute values.

*•Please convert to mm/s which is more widely used for DV than mm/h & agrees with the units used on the other plot on which you have DV.*

Authors response: The units for DV are mm/s everywhere. There was a mistake in the figure caption. Corrected.

**Referee #2's comments:**

*The manuscript presents component analysis of the impact of urban areas on air quality in Euorpean cities using several scenario simulations in a regional offline-coupled model system. The manuscript is well written and easy to understand. However there are some issues to be addressed, including the flow and the structure of the manusciprt that can be found below.*

*Comments:*

*Section 2.2.*

*Are fire emissions not taken into account? They can be important episodically for the O3 levels.*

Authors response: Fire emissions are not taken into account. We admit that wildfires can significantly contribute to total pollution (not only PM but also gaseous pollutant e.g.  NOx and CO) and model biases tend to improve if such emissions are included (Lazaridis et al., 2008), their contribution is important rather over southern Europe and Mediterranean and not on our focus area (central Europe). Moreover, we were interested (among other urbanization impacts) in the impact of urban emissions which not normally include wildfire emissions.  We made a short note in the manuscript concerning these emissions with the above argument why we omitted them.

*A comparison of the two CAMx versions can be provided in the supplement as difference in the chemical mechanism can have significant impacts on the pollutants considered in the present study? Similar results could point the emissions however different results point both emissions and chemistry. I see this has been discussed in the last section but I think it should also take place, at least partly, here as it now stands detached from each other.*

Authors response: We think it would go far beyond the scope of the manuscript to include a graphical comparison (using plots) of two different CAMx versions, although such comparison could be, in general, be very useful. We however added a few further notes on the difference between CAMx v 7.10 vs. 6.50. to the "Models used" section. As emissions in this study were the same as in our previous similarly oriented studies (Huszar et al.2020b, 2021) and the only difference is in the newer CAMx version, the differences in model performance can be explained by model version differences. In this sense, the improvements in CB6r5 vs. to older CB5 are certainly the most important and stand behind the reduction of the model biases. This is detailed in the Discussion section (2$^{nd}$ paragraph) with highlighted improvements in this new mechanism with respect to CB5.

*Section 3.1*

*The plots in Figure 2 are representing cities in the different domains or are they all from the 9 km mother domains? What is the driving meteorology in these plots, WRF or RegCM? Is there a comparison available for the meteorology and associated chemistry over the 9 km grid?*

Authors response: Results for Prague are taken from the 1 km domain, otherwise they are extracted from the 9 km one. This is made clear in the revised manuscript. In the original "Discussion" version of the manuscript, the driving meteorology was RegCM, but based on the other Referee's comments, we decided to include the plots also for the WRF driven CAMx runs (here all cities are represented at 9 km x 9 km). This allows us to see what effect has if an alternative driving meteorological model is used. Moreover the 9 km results serve as a good justification that even at this regional scale resolution, the model performance can be acceptable and the concentrations well represented (except SO2, but this is probably not a resolution issue). Finally, we also included another column to the validation figure showing the diurnal cycles of ozone, as for this pollutant, monthly values are no so policy relevant and the hourly evolution is indeed important (especially the daily maxima). As for the meteorology, we rely on our previous study of Huszar et al. (2020b) which used exactly the same meteorological driving data (from both RegCM and WRF) and provides the validation of the meteorological fields: namely near-surface temperature, precipitation, 10 m wind speed and PBL height, which are all important from air-quality perspective.

*Section 3.2*

*Does Figure 3 provide the ensemble from both the WRF and RegCM simulations? Please explain and modify the caption accordingly.*

Authors response: yes, the results represent the ensemble from both simulations. This is clarified in the manuscript.

*Section 3.3.4*

*Can you please a bit more the case for Milan as it really stands out?*

Authors response: Milan, Italy has one of the highest emissions among cities and also the worst air quality (https://www.eea.europa.eu/themes/air/urban-air-quality/european-city-air-quality-viewer) which is captured also by the model (see e.g. the validation figure with monthly/daily cycles for NO2). Furthermore it has the warmest climate among the chosen cities and a relatively large size, so it is clear the the urbanization causes strong changes in BVOC emissions, see the figure with BVOC emissions changes (for Isoprene in Fig. 9). Strong decreases of ozone, as detailed in the Discussion, are then result of decreased VOC emissions in a VOC-limited chemical regime. The strong secondary responses of NO2 are the result of these strong ozone changes. We added a note to this in the revised manuscript.

*Can you also elaborate a bit on the impact of resolution on these results focusing on the Prague experiment? How much your conclusions would change based on this experiment if you were able to run all the cities on 1 km resolution for example? I am aware this cannot be answered quantitatively without the simulations, but I would like to see a discussion on this.*

Authors response: In this regard our discussion should be based especially on the comparison of results for Prague for 1 km vs 9 km resolution. Such comparison shows that the impacts are stronger for the 1 km resolution peaking at the city center, i.e. the largest impact are concentrated in the city core while at 9 km resolution the impact is logically uniformly distributed within the 9 km gridcell, so the peaks are flattened.  This is very well seen in each of the four contributors. It can be thus assumed, that had the model been run at 1 km  x 1 km resolution for all analyzed cities, the impact would have had a much larger peaks (in absolute sense) concentrated around the cities cores. We added these notes to the revised manuscript (at the Discussion).

*Section 3.4 focuses on explaining the diurnal variations but do not discuss much the underlying reasons for these diurnal variabilities. I would expect such a discussion supported with some plots, likely in the supplement. I see these decoupled explanations also in other parts of the manuscript. There is of course not a correct way to provide this information but I think the manuscript would benefit very much if the discussions in the last section could be moved to the corresponding sections explaining the impacts of the different scenarios.*

Authors response: We wanted to keep the logical practice that the Result section provides the results without searching for explanations or without deeper discussions, while it is the Discussion section were different findings are explained, interpreted and interconnections are formulated. We would prefer thus to keep the Results section "clean" from deeper discussion. For the diurnal patterns of different contributors, we added a new paragraph in the Discussion section, which explained the modelled cycles and reflect them to findings in previous studies.

References (not used in the original manuscript before the revision):

Lazaridis, M., Latos, M., Aleksandropoulou, V. et al. Contribution of forest fire emissions to atmospheric pollution in Greece. Air Qual Atmos Health 1, 143–158. **https://doi.org/10.1007/s11869-008-0020-0**, 2008.

---

## Author Response (AR2)

**Authors responses (1st and 2nd round) on the Anonymous Referee #1 review of "Impact of urbanization on gas-phase pollutant concentrations: a regional scale, model based analysis of the contributing factors"**

by Huszar et al. (acp-2022-337)

We thank anonymous referee #1 for his very detailed review and all the comments. We will address each of them and our point-by-point responses follow below. Reviewer's comments are *italicized*.

Our 2nd round answers are written in **bold**.

The authors investigate the impacts of some key assumptions about representation of urban anthropogenic emissions, dry dep and biogenic emissions, and meteorology from the urban canopy on the fine scale representation of ozone, SO2, and NO2 across a part of Europe with a lot of cities using an uncoupled modeling frameworks employing offline meteorology from two different regional climate models. They look at winter and summer averages for SO2 and NO2, and summer MDA8 ozone.

This is a useful contribution to the peer reviewed literature, but the framing of the paper and its implications need work to get the paper to be publication ready.

Major issues.

The authors sometimes overly generalize (& in some cases w/ excessively strong statements).

For example,

•I find that the use of the term 'the urban canopy meteorological forcing' is too general. It's hard to imagine how the authors can say that the impact of the urban canopy meteorological forcing' always decreases primary pollutants but increases ozone. to me, this is equivalent to saying: 'meteorology always increases ozone'.

•The statements "these are the two major drivers of urban air pollution", "the two minor contributors can be neglected" & "it is clear that the main driver affecting..." are quite strong... I don't believe they should be this strong, based on the evidence (or lack thereof) presented by this work. The authors are testing very specific assumptions about different processes.

•The discussion of local versus rural/other cities' influences on air pollution in the Intro seems limited. some pollutants' distributions and production are very regional scale, so separating sources into local vs. rural doesn't make much sense. Additionally, it seems strong to say that 'air pollution in cities is mainly determined by local sources'... for ozone, background levels in cities should be an important fraction of the total ozone.

Authors response (1st round): The term "urban canopy meteorological forcing" (UCMF) was introduced and defined in Huszar et al. (2020) as the forcing that the urban canopy has on the city scale meteorological conditions and thus on air-chemistry via modified transport, increased turbulence, increased temperatures etc. In other words, it is the urbanization induced changes of meteorological conditions. If we thus refer to the impact of UCMF, we do not refer to the impact of urban meteorology as it is, but the impact of the urbanization induced changes only. In Huszar et al. (2018a,b) and in many other cited literature it is shown that one of the most important components of UCMF is the increased turbulence which causes decrease of primary pollutants overweighting the decreased wind speeds and reduced dilution (as another component of UCMF). On the other hand, for ozone, which is a secondary pollutant, the effect of UCMF is more complicated as UCMF acts on both ozone itself and its precursors. The net result in this case is an increase with strong contribution of decreases NOx concentrations and reduced titration (see Huszar et al., 2018a). Consequently, we can indeed say that UCMF increases ozone. We however admit that we cannot write "secondary pollutant" in general, as this impact

can be different for different secondary pollutants (e.g. Huszar et al.,2018b showed decreases of secondary inorganic aerosol due to UCMF), so we modified the text to be more specific.

Regarding the strong statement of "These are the two major drivers of urban air pollution"... we refer here to the fact the if we consider the total impact of urbanization (i.e. the rural to urban transformation) on air pollution, we can clearly state that the two most important factor that constitutes this impact are the urban emissions themselves and the UCMF. We are not stating here that e.g. the background state or rural emissions do not play role in urban air pollution. We are only interested in the role of urbanization (rural emissions and the background air pollution is the reference base state), i.e. if a rural area is urbanized, its air pollution is significantly modified and the most important factors in this modification are the emissions and UCMF. In the revised manuscript, we rephrased some of the above criticized statements to be more precise.

Regarding the statements about the two minors contributors, we admit that although their contribution is much smaller, based on this work we cannot state that they can be neglected. This is indeed a strong statement and we removed it from the text (and rephrased the remaining text slightly).

As for the *"The discussion of local versus rural/other cities' influences on air pollution"*, we have to stress, that the purpose of the analysis was to evaluate how urbanization affects the local air pollution, i.e. air pollution over the area where the urbanization occurs. The study is neither focusing on i) how urbanization affects the regional scale air pollution (although some results are obtained in this regard too) nor on ii) how the rural/nourban areas affect the urban air pollution. This was also specified more clearly in the manuscript.

Authors response(2st round): Yes, we admit that the urban canopy meteorological forcing (UCMF) is a general term. As we defined it with in Huszar et al. (2020a), it is a the change of the meteorological conditions due to the introduction of urbanized landsurface (i.e. buildings, streets, specific material properties covering them and anthropogenic heat). However, UCMF has very specific effects in terms of individual physical quantities: increased temperatures, reduced city-scale winds, increased vertical eddy-diffusion, reduced absolute humidity and so on. So UCMF is a general term encompassing well known modifications in meteorological conditions. So if we refer to the effect of UCMF, we refer to these modifications and not the effect of the "absolute" meteorological conditions in cities. The question of what these meteorological modifications have on pollutant concentrations has been posed and answered in many previous studies including studies we published earlier (Huszar et al, 2018a,b; Huszar et al., 2020a; Huszar et al., 2021) and numerous other studies referenced in. As already pointed out in our  $\mathbf{1}^{st}$  round answers above, these studies showed that the dominating component of UCMF is the increased vertical eddy-diffusion and it indeed results in stronger vertical removal of primary pollutants from the urban canopy layer which leads to their decreased concentrations. On the other hand, for ozone as a secondary pollutant, the effect of UCMF depends on its effect on primary precursors along the with direct effect on ozone. Here, our and other previous works have shown that UCMF results in increase of O3 as the results of stronger removal of NOx and smaller titration of ozone. These results are to be viewed as the time averaged impact of UCMF, the instantaneous impact can be of course different depending on the actual meteorological conditions. So, to say "UCMF has some effect on pollutant X/Y" is not the same as "meteorology has that effect on X/Y", because UCMF is a modification and we are interested in how this modification propagates to pollutant concentration. We understand that this can be misleading if the reader is first faced with UCMF and the referee is right that it can be though on the wrong way that the "meteorology decreases primary pollutants and increases ozone". We stressed in the revised manuscript, that UCMF is a modification of meteorological conditions and we are interested in how this modification propagates to modification of pollutant concentrations.

Regarding the statements about the "major" and "minor" drivers of urban air pollution, we meant the relative importance of different processes controlling urban air pollution. More specifically, we were referring components of the rural-to-urban transformation (encompassing 1) change of rural land-surfae

to urban one and 2) introduction of urban emissions), which means four different contributors defined in our study. From these contributors, we found that the effect of urban emissions and the meteorological changes introduced by urban land-surface are the strongest while the other two has a much smaller magnitude. These are however our results and do we not want to generalize (we can imagine of a city, were e.g. the "DBVOC" effect can be very strong comparable to anthropogenic emissions or UCMF). We thus stress in the final conclusions that these results are for central European climate conditions and typical "rural" land-cover in this geographic region.

Regarding the discussion of the "effect of rural/other cities effect", as the referee suggest, we added a short paragraph on this issue in the Intro. We agree with the referee that for e.g. ozone, the regional scale air pollution is very important with respect to local/urban scale concentrations. However, as we already said in the 1st round response, the purpose of the study was to evaluate how urbanization affects the local air pollution, i.e. air pollution over the area where the urbanization occurs. More specifically, we were interested in "what happens if I put a city in this rural area?How the pollutant concentration of the area changes and what are the underlying processes and their magnitude?". In our study we wanted to show that for a comprehensive picture of the resulting air pollution, we must consider not only some additional emissions that this brings in, but also the modified meteorological effects (UCMF) and changes in dry-deposition and BVOC emissions. It is clear, that the final pollutant concentration will be affected by the background air pollution or by the effects of other cities, however this is not the focus of the study. We are preparing a study were we will specifically focus on how rural emissions (both anthropogenic, biogenic and natural like dust) and long-range transport affect the urban scale air pollution. This is an interesting and important topic deserving a separate study.

**There are scattered introductions to urban air pollution and ozone chemistry throughout the manuscript. It would be more helpful if this was concentrated in the Intro.**

Authors response(1st round): Some basic information on urban gas-phase air pollution (the three analyzed pollutants) is already provided in the Intro, but we wanted to keep the Intro short focusing rather on collecting the current knowledge about how urbanization affects the air-quality and what potential contributors it has. We admit, that some information regarding air pollution and ozone chemistry appears also elsewhere in the manuscript, but it was necessary in the context of that text, especially in the Discussion, where many of the modelled features and their interpretation required it.

**Authors response(2st round): We added some more information on how urban emissions influence urban gas-phase air-pollution (distinguishing between primary and secondary pollutants) and as said in the 1st round of our response, we want to keep some information regarding urban gas-phase air pollution elsewhere in the text too as it was necessary in the context (mainly in the Discussion).**

With respect to the introduction of the simulations in the Intro, it is not clear what the authors are investigating here. Are the authors using a global model? What is the resolution of the model? What is the base simulation? Are the authors assuming a preindustrial like state? Because this is a model study, it seems like this all should be very clear in the Intro.

Authors response(1st round): We admit that we could do more in the Intro to make clear, what we are investigating, what is the reference state etc. We added some text to clarify this. E.g. we stressed that the purpose of the study is to evaluate the i) total impact of urbanization (``DTOT") and more importantly, the contribution of ii) each of the urbanization related impacts (i.e. ``DEMIS", ``DMET", ``DLU\_D" and ``DBVOC") over regional scale domain on the present day final urban air pollution levels using coupled regional climate and chemistry transport models applied at moderate horizontal resolution (9 km), so we stressed the facts that 1) regional scale models are used, for 2) present day conditions (including emissions and landuse), 3) the resolution used. We also added a sentence explicitly saying to which reference state we consider the effect of urbanization, i.e. the state "without cities", which means the urban landuse replaced by rural one and urban

emissions removed (the way this is done is described in the Methodology and within an answer to the referees comment further below).

Authors response(2nd round): In the revised manuscript we included information to answer all the questions above. i.e. we do not use global model but a regional scale climate and chemistry transport model(s). The global model is neither serving as a driver for meteorology, instead we use reanalysis to drive the regional climate. For the chemistry transport simulations, the CAM-chem global model is used, but we did not run this model, only used its data as chemical boundary conditions. We also clearly defined the background (reference or base) state in the revised manuscript, i.e. state "without cities", which means the urban landuse replaced by rural one and urban emissions removed (as already said in our 1st round response). So we cannot say the we assume preindustrial state as reference, as this would mean also to assume preindustrial climate (GHG concentriations) and preindustrial large scale chemical background. Instead, we assumed current climate and large scale air pollution and we investigated the total urbanization effect (as as we call it RUT) within this environment. We made this even more clear in the manuscript.

[Later in the methods, the authors say: "in the first experiment where urban emissions are disregarded, we removed urban emissions only for the 19 cities chosen for the analysis". I think my fundamental question is: how are the authors defining what emissions are "urban"? what exactly are they investigating with this first simulation? This should be clear in the Intro, as well as the methods.]

Authors response(1st round): this is indeed a fundamental question and we detailed in the Methodology section but also put a note in the Introduction, that: the urban emissions are defined as those falling within the city administrative boundaries (based on the ISO 3166 standard defining *countries and their subdivisions*.). The emission model we used (FUME) allows to mask out emissions based on a given geometry (shape), i.e. we could remove the emissions from all gridcells which completely lie within the geometry, while from those gridcells, across which the city boundary goes, only that part is made zero, which falls within.

The first simulation with CAMx (driven either with RegCM or WRF) is meant to be the reference (base) state, while with each further simulation we added one effect. The first effect added are the urban emissions, so the first two experiments allows us to evaluate the effect of urban emissions only (while the 2nd and 3rd allows to evaluate the effect of landuse on dry-dep, and so on, as defined by Eq. 4 to 8).

Authors response(2nd round): As said in our 1st round response, urban emissions were distinguished from rural emissions purely geographically – more specifically all emissions were "urban" which fall within the city's administrative boundaries (in the manuscript, we explicitly mention what we mean administrative boundary). With the first simulation, we create the model representation of the reference state, which mean rural landuse for all processes/models (meteorological model, dry-dep and biogenic emissions model) and urban emissions removed (as stated above).

How is the land use input required for the regional climate models similar/different from the input to the dry dep and biogenic emissions schemes? Seems like the authors need to emphasize that in some of their sensitivity simulations, the deposition/biogenic emissions are forced by meteorology that is decoupled from the land use type impacting the deposition/biogenic emissions.

Authors response(1st round): The data for landuse information is the CORINE CLC 2012 landcover data (https://land.copernicus.eu/pan-european/corine-land-cover) which was used within RegCM and WRF as well as for the dry-deposition scheme in CAMx. As CORINE does not contain information needed for MEGAN, namely the annual cycle of the leaf-area-index (LAI), the plant functional types (PFT) and the plant emission potentials, these information were derived independently based on the data provided in Sindelarova et al.(2014, 2022). The meteorology driving the MEGAN model is either the one considering NOURBAN (rural) landuse, i.e. without the UCMF effects, or, the URBAN one, i.e. which does considers URBAN landuse (with the applied urban canopy

model), see Tab. 1. (two pairs of RCM runs – with RegCM and with WRF). With respect to the effect of urbanization on biogenic emissions and the consequent effects on air-quality, three experiments were important: No. 3, 4 and 5 (Tab. 2.). In experiment 3, we considered rural land information for MEGAN and NOURBAN (rural) meteorology, so in this experiment, biogenic emissions represent the "nourbanized" situation (i.e. the base/background state). On the other hand, in experiment 5, biogenic emissions are driven with urban meteorology (i.e. which considers urban effects – UCMF) and considers urban land information (reduced-zero LAI in our case). This experiment thus represents the full urbanized case concerning the biogenic emissions. To examine the partial effects of the modified land information and the UCMF on biogenic emissions, we performed the experiment 4, where the land information for MEGAN was "urban" while the meteorological driving fields were those considering NOURBAN (rural) landuse. We tried to emphasis this more clearly in the revised manuscript.

Authors response(2nd round): Indeed, as the referee says, we performed simulations were the landuse used for dry-dep is decoupled from the landuse used for the meteorological driver (RegCM or WRF). This was necessary to evaluate the partial impact of modified dry-deposition on deposition velocities and concentrations. Also this is the reason, why we chosen to use an offline couple, which enabled us to use rural landuse for meteorology (in RegCM or CAMx) and urban for dry-dep in CAMx. The same is true for the effect of BVOC, which are a result of two effects, the urbanization induced vegetation changes and the modified meteorology driving these emissions. We wanted to decompose the BVOC emissions changes into these two parts and this required to run the emissions model with rural meteorology and consider urban landuse at the same time, or vice-versa. The revised manuscript stresses this.

The approach used by the authors to separate the roles of urban emissions, meteorology, dry dep, and BVOCs (detailed at the end of page 8) is full of assumptions. The effects are not additive; the authors probably should not assume equation (2), although generally looking at the difference between simulations to gauge sensitivity is fine. The authors need to adjust the framing of the work on this front (e.g., "individual components" – I wouldn't characterize as individual components when the components are tightly coupled in real life, and should be in the model).

Authors response(1st round): as we said, the purpose of the study was to quantify the individual contributors to the total impact of urbanization as well as the total impact itself (DTOT). The total impact for a species i (expressed as its average concentration ci) is calculated as the "full" (real, i.e. urbanized) situation minus the background (base, rural, reference) situation denoted  $c_{i,rural}$ . What we call the the total impact of urbanization is the difference,  $c_i - c_{i,rural} = \Delta c_{i,RUT}$ .(RUT - rural to urban transformation). This is our equation (1). As we were interested in decomposing the  $\Delta c_{i,RUT}$  into different parts, we performed additional simulations in which we added different effects one-after-other as defined in Tab 2. And the effects themselves can be calculated as written in Eq. 4 to 8. Indeed, if we add these four equations (5-8), we will get the right hand side of Eq 2. and the EUYuuU(6) – ENNrrN(1) difference. The later is exactly the total impact of urbanization of one of the contributors. So, the way how the effects are calculated makes them mathematically perfectly additive, as they are not added individually (separately) to the base simulation, but added gradually (in a cascading way). In other words, we have 6 simulations, SIM1, SIM2, SIM3, SIM4, SIM5 and SIM6, and what we state by equation 2 is that the Eq. SIM6-SIM1 = (SIM2-SIM1)+(SIM6-SIM5)+(SIM3-SIM2)+(SIM5-SIM3) holds, which is trivial.

We tried to clarify this more clearly in the revised manuscript and instead of "components" we will call them "contributors to RUT".

Authors response(2nd round): The main goal of the manuscript was to quantify the total impact of urbanization (RUT), but also we wanted to somehow quantify the roles of the underlying causes, which we defined four: the effect of modified emissions (adding urban emissions), the effect of UCMF, the effect of modified landuse on dry-dep and finally the effect of BVOC emissions changes. This is often

the case with the estimation of other anthropogenic impacts, e.g. many studies looked at only the direct radiative impact of aerosols ignoring the indirect effects, while it is clear, the that the two effects act simultaneously and are not additive. The question for us was how to handle this problem, how to attribute some numbers to the role of the four mentioned effects which act simultanously.

Basically, two approaches were possible.

a) we could make the "full" urbanized simulation were all urban effects, urban emissions and urban landuse is considered (i.e. the simulation EUYuuU numbered 6 in our setup) and then to remove individual effects from this "full" simulation (the so called annihilation method), but always to remove only one effect. With this approach, we would get the magnitude of each effect. We also would make one reference simulation were rural landuse, no urban emissions were considered (simulation ENNrrN numbered 1 in our setup). With this setup however, and this is due to non-linear feedbacks, it is clear, that the total impact of urbanization calculated as EUYuuU- ENNrrN (which is of course same as in our setup) would not be the same as the sum of individual impact, so they would not be additive.

b) Instead of a), we could follow a slightly different approach, and it is a stepwise addition of effects oneby-one to the reference simulation to end up with the real case one. Lets call the simulations as follows:
1) SIMref, - reference simulation with no cities/urban emissions

2) SIMref+emis – simulation with urban emissions added (but landuse is "rural" for all processes)

3) SIMref+emis+LU\_D – simulation with urban emissions added and urban landuse for dry-dep (the landuse for BVOC emissions is still rural)

4) SIMref+emis+LU\_D+BVOC – simulation with urban emissions added and urban landuse for dry-dep and BVOC emissions

5)  $SIM_{ref+emis+LU_D+BVOC+UCMF}$  – simulation with urban emissions added, urban landuse for both dry-dep and BVOC and urban landuse for meteorology (UCMF considered), i.e. this is the "full" (real) case.

Let's denote some average concentration (e.g. DJF average  $NO_2$ ) for a chosen pollutant  $C_i$  where the index "i" is one of 1-5. In our set-up, we calculated the individual effects called by us as "contributors" as follows (for the chosen pollutant):

 $\mathbf{C}_{\mathsf{DEMIS}} = \mathbf{c}_2 - \mathbf{c}_1.$

 $\mathbf{C}_{\mathsf{DLU}_{\mathsf{D}}} = \mathbf{c}_3 \mathbf{-} \mathbf{c}_2$

 $\mathbf{C}_{\mathsf{DBVOC}} = \mathbf{c}_4 - \mathbf{c}_3$

 $\mathbf{C}_{\mathsf{DMET}} = \mathbf{c}_5 - \mathbf{c}_4$

 $C_{\text{DTOT}} = c_5 - c_1$

Thus we can write (by simply adding the first four equations):

 $C_{\text{DEMIS}} + C_{\text{DLU}_D} + C_{\text{DBVOC}} + C_{\text{DMET}} = c_2 - c_1 + c_3 - c_2 + c_4 - c_3 + c_5 - c_4 = c_5 - c_1 = C_{\text{DTOT}}$

This is exactly the same as stated by equation 2) in the manuscript, and means that with this way of quantifying the impacts (DEMIS, DLU\_D, DBVOC, DMET and DTOT) they are additive. But we have to stress that they are additive within our simulations and in the real life they act simultaneously and due to nonlinear feedbacks additivity does not hold, it is only an approximation. We stressed this in the revised manuscript.

For ozone, all the cities have very similar observed annual cycles for urban areas, and the model captures them. I'm not sure this is the best way to evaluate ozone for the purposes of this paper. The annual cycle is likely largely driven by regional scale phenomena rather than urban processes. It would be good to evaluate whether the model captures urban-suburban-rural gradients in ozone.

Authors response(1st round): We admit that monthly means of ozone are not very representative in terms of model performance and the model's capability to resolve the hour-to-hour variation is important for this pollutant. Therefor we added the average summer diurnal cycle of the modelled and measured values as a further column in Fig. 2.

Authors response(2nd round): The study looks at urban concentrations of gas-phase pollutants and their modifications due to urbanization (and its contributors). So we wanted to limit our validation to urban observations only. We also agree that monthly means of ozone are driven largely by synoptic scale processes over regional scales so this is why we added in the revised manuscript their average hour to hour variation as diurnal cycles (for summer months). These plots (3rd column of the revised "validation" figure) clearly show that average daily maximum ozone values are not very large, which is in accordance with the expectation that over urban areas, local minima of ozone pollution is encountered due to strong NOx emissions, NO titration and overally a VOC-poor environment. We saw in our simulations we saw that the rural ozone values are much larger resulting in a strong rural-urban ozone gradient, however these are not presented as not the focus of the study.

**Also, I wouldn't call an uncoupled modeling framework a novel approach.**

Authors response(1st round): the "novelty" of our approach was in decomposing the total impact of urbanization into individual contributions and this required often to decouple different effects from meteorology/landuse. This was not possible to do with a online coupled model, so the fact that the models (including the biogenic emissions mode) were offline coupled to the chemistry transport model was an necessity. However, we removed the word "novel" in our Introduction to avoid doubts.

Authors response(2nd round): As said in our first response, the nature of our experiments where different processes (dry-dep, BVOC emissions) had to be driven by different landuse and/or meteorology (i.e. landuse effects and meteorology were decoupled) made it necessary to use a modelling system, were different components: meteorological driver (i.e. the regional climate model), the biogenic emissions model and the chemistry transport model had to be run separately, i.e. using offline coupling. For example, if we had used WRF-Chem, we would not be able to use different landuse for the atmospheric part and different for biogenic emission module. Also the landuse for dry-dep has to be same as for other components of WRF-Chem. Using offline couple of a RCMs (RegCM or WRF), a biogenic model (MEGAN) and a chemistry transport model (CAMx) made it possible to separate the individual effects and to fulfill the goal of the study. But again said, we rather removed the world "novel".

**Minor issues.**

There is a fair amount of grammar issues that a closer read or English editing could help (I didn't list these).

Authors response: we revised our manuscript in this regard and corrected it with the help of an English editor.

**Other minor comments.**

Intro.

•What do the authors mean an 'artificial' surface?

Authors response: This means not natural surface, i.e. surface with artificial objects like buildings, roads, parking etc.

•Ref for 60% of the population will live in urban areas

Authors response: Reference added (UN(2018b))

•The authors shouldn't start a sentence with 'E.g.'

Authors response: Corrected.

•Sure, plants are a large sink of many gases, but that doesn't mean other surfaces can't be either. What's known about dry deposition to urban surfaces? Do we as a community know what deposition velocities should be over urban surfaces?

Authors response: indeed, deposition to urban type landsurface in regional models is a complicated issue as urbanized surfaces are comprised by very different materials and the vegetation fraction of urban areas is also highly variable. The used dry deposition model (Zhang, 2003) considers some average value but we admit that this might be burdened by a high uncertainty given the fact that substantial differences between cities exist. Nevertheless, in their model, urban surface has a clearly smaller capability to remove gaseous material from the air compared to purely natural surface. However, some secondary pollutants like HNO3, HNO4, H2SO4, H2O2, HONO or also NH3, the removal in case of wet canopies is higher for urban areas than for rural ones (e.g. crops) due to their high solubility and reactivity with solid surfaces. So in general, urbanization modifies the removal depending the the pollutant itself and the state of the canopy.

•Again, yes, plants are a large source of BVOCs, but that doesn't mean that the plants that we do have in the cities aren't very efficient producers of BVOCs, or there aren't emissions of BVOCs from consumer products. (In the past couple of years, there has been evidence of both. To be clear, I'm not asking the authors to investigate efficient emissions of BVOCs from urban vegetation or BVOCs from consumer products, rather contextualize and motivate their study accurately/well)

Authors response: indeed, urban vegetation like any other vegetation is a great producer of BVOC but when the rural surface is turned into urban one (during the RUT) than this vegetation is reduced (will not be zero, only reduced) and consequently the emissions of biogenic gases are also reduced. In the revised manuscript, we further added a note on the VOCs from consumer products that are normally of biogenic origin (e.g. isoprene and monoterpenes) to make the reader aware that reduction occurs due to reduced vegetation but increase occurs due to some anthropogenic input.

•Use of 'background' as defined as 'without urbanization' is not great, given there is clear definition of 'background ozone' (https://doi.org/10.1525/elementa.309)

Authors response: we agree that "background ozone" has a well defined terminology, however we had to define a background (**reference**, base) state here to which the individual urban effects are gradually added, therefor we prefer to use "background" for the non-urbanized situation, **but to make it clear to the reader, we call it alternatively as "reference" and added a note to not to confuse with "background ozone" which a well defined term.**

**Methods**

•The authors are taking grid-cell average meteorology and feeding the CTM with this? Needs to be super clear.

Authors response: the meteorology provided by the driving RCM (RegCM or WRF) is defined in points – in case of state variables (temperature, humidity, cloud/rain water) these are the grid-cell centers, in case of wind

components and diffusion coefficients these are defined at grid-cell faces. The state variables represent grid-cell averages in CAMx, the same for the concentrations. This was clarified in the revised manuscript.

•What is the point of discussing CLMU in CLM? Is CLM coupled to RegCM4.7? Also, what do the authors mean by 'the traditional urban geometry approach is implemented'?

Authors response: we admit that in the current formulation, the reader could have the impression that CLM4.5 (in which CLMU is included) is a separate model independent from RegCM, which is of course not true. CLM4.5 is the surface exchange model implemented in RegCM4.7 (replacing the older BATS surface scheme). We reformulated the sentence to clarify this.

•It seems like it should be emphasized that the authors are looking at not only two different regional climate models, but additionally two different urban canopy parameterizations.

Authors response: yes, this is true and we stressed this in the "Model simulations" section.

•Do S&P 1998 and Zhang 2003 give wet deposition parameterizations, or just dry dep? Also, it seems like the authors should elaborate on how urban dry dep is simulated in CAMx and its key sensitivities.

Authors response: S&P stands for the wet-deposition while Zhang 2003 refers to the dry-dep scheme. This was clarified in the revised manuscript. We also added some key information how the deposition velocities are calculated and what resistances are considered as well as some notes on how these are modified if urban landsurface is considered and on what meteorological parameters are they dependent.

**•How do the authors know that the impact of soil NOx is small?**

Authors response: In our experiments to soil-NOx emissions are about two orders of magnitude smaller compared to the BVOC emissions (we plotted these emissions but not presented in the manuscript). Therefore we expected their effect to be much smaller, as well as the effect of their urbanization induced modifications. We made a note on this in the revised manuscript.

**•What does 'showed that their long-term effect is rather small' mean?**

Authors response: We meant that the long-term (10yr in the mentioned study) radiative effects of urban pollutant emissions (i.e. not GHG emissions) is very small and is comprised mainly by the cooling effect of aerosol, while the effect of ozone modifications due to urban emissions are almost negligible. This was clarified in the revised manuscript.

**•Clarify that the nested domains are only for one of the regional climate models**

Authors response: This is later clarified in the Model simulation section.

•What does 'they found a rather small impact' mean (end of first paragraph of 2.2)?

Authors response: we extracted some more specific, quantitative information from the mentioned studies, i.e. that the sensitivity of ozone concentrations to model resolution is about 10% with higher model resolutions producing smaller concentrations. **Clarified in the revised manuscript**.

•What does 'brings some accounting for the uncertainty related to the urban land-cover representation' mean?

Authors response: it means that due to the fact that the landuse is represented differently in WRF and RegCM, dominant landuse category vs. fractional, respectively, partly urbanized surfaces can be differently accounted for in the two models: for example. if 60% of the gridcell is covered by urban surface than in WRF (dominant landuse), the total gridcell area is considered as "urban", while in RegCM only the 60% of the gridcell (fractional). This means, that some of the urban meteorological effects can be stronger (or different in more general fashion) in WRF. We added a note on this to make clear what we meant.

•How do they authors grid the irregularly shaped Czech emissions to the model? Is this where FUME comes in? If so, I'm confused why the descriptions are in separate paragraphs.

Authors response: all emissions all interpolated by FUME, so the regular lat-lon CAMS data as well as the irregular Czech emissions. The reason is that the model grid is always different from the input data spatial distribution, regardless of the fact if these are regularly gridded or irregular shapes. In the revised manuscript, we reorganized the Paragraphs. Now all the information regarding anthropogenic emissions including FUME are in one paragraph.

•*I'm* confused when the authors are discussing the "city" vs. "non-city" portions of the cross-boundary shapes. Is there a sub-grid distribution of emissions?

Authors response: we meant that emission over emissions cells (regular in CAMS or irregular in the Czech emissions), which are divided by the boundary of the city, has to be split into the part that lies inside the city boundaries and the part lying outside. This is were the masking capability of FUME can be used. **This has been made clear in the revised manuscript.**

•In replacing the urban land with the rural crop land over the entire domain (not just the cities chosen), the authors may be changing background levels of air pollutants, not just local urban levels.

Authors response: it is true that urban land use was replaced to rural one (crop) over the entire domain, but the effect on background levels is expected to be small for two reasons: 1) emissions from these areas were still considered, 2) the urban meteorological effects from these (minor) urban areas has rather small influence or air pollutants as the UCMF is also small (see e.g. Huszar et al.(2014) so the effects are important mainly around large urban centres we are interested in also in this manuscript). We included this argument in the revised manuscript.

**Results**

•What's the first difference? I only see a reference to the second difference in the validation section. Also, if a different chemical mechanism is used, then that seems like a big change.

Authors response: the first difference was the newer model versions. We reformulated this paragraph to more stress that there are indeed differences so some level of validation is necessary.

•I don't know what the authors mean by "while all urban and suburban background stations were used from a subset of the analyzed cities". Why a subset? What does background mean here? Which sites were chosen for the evaluation & why? Needs to be clear.

Authors response: we agree that optimal would have been to choose station for all urban areas. However we made this subselection in order to reduce the amount of subfigures. Further, instead of an average-over-allcities, we rather wanted to show whether there are some notable differences between individual cities or the model biases/performance is similar over different city. Our selection of cities for validation (8 from the total of 19) followed the criteria being largest from the total of 19 cities chosen. Another criteria was to select cities which are more-or-less evenly distributed over the domain, so the validation conclusions from them can be applied to other cities. Lastly, we had to select cities with sufficient number of measuring stations (which correlates usually with the city size of course.) Regarding the selection of the station, we chose AirQualityStationArea = "urban" and "suburban", while only AirQualityStationType = "background" for the stations to fullfill. "Background" stations were needed for representativeness of the concentrations over a larger area (like gridcell). In the revised manuscript, we explicitly say according to which criteria we selected the subset and the stations.

•It seems like there should be some text about the diel cycle scaling of the urban emissions in the methods.

Authors response: we provide a reference for the temporal dissaggregation factors used to calculate hourly emissions from annual one (van der Gon et al.(2011)). These include a sector-specific diel cycle to obtain hourly values from daily means (which are obtained using monthly factors and factors for week-days). As these are dependent only on the activity sector but not on the geographic location, they will be certainly not equally representative for all cities which could add some biases for some of them. We made a note about the temporal factors in the Method section.

**Discussion and conclusion**

•Word choice -- the authors selected four contributors based on previous work to investigate, they did not identify them based on their work ...

Authors response: we rephrased the sentence and removed the "identify".

•A lot of the discussion, especially with respect to model evaluation, is framed as a comparison to the authors' previous work. This is not too helpful for the reader who is not familiar with that work. Can the authors focus on what knowledge is generated from this work first, and then compare (briefly) to previous work by the same author? In other words, the comparison should be secondary, not primary.

Authors response: we reorganized this paragraph to stress what "knowledge was generated" at first place and than how this knowledge relates to previous work or how it can be explained in the context of previous work. This was sometimes necessary as our experiments evolved from these and the updates in the experiments usually explained many of the model performance improvements.

•I'm not sure how the authors can go from "in the case of SO2, the model is rather unable to correctly resolve the annual cycle of near-surface concentrations" to "in summary, we did not identify substantial model biases" within a couple of sentences...

Authors response: we agree with the reviewer and removed this formulation.

•I recommend cutting the "hints that the effect of urban emissions is well captured" part

Authors response: removed.

•Why is there stronger dry dep due to higher temperature? If there are meteorological sensitivities to the biogenic emissions and dry dep parameterizations, then they should be spelled out in the descriptions of the models.

Authors response: Here we refer to Huszar et al. (2018a) who writes "In the dry-deposition scheme used (Zhang et al., 2003), the aerodynamic resistance ( $r_a$ ) and the stomatal resistance ( $r_s$ ) depend on temperature. The first is calculated according the near surface stability and mixing conditions following the Louis (1979) scheme incorporated within CAMx. Higher near surface temperatures result in reduced stability decreasing the corresponding resistance. The stomatal resistance decreases strongly with temperature as a result of more opened stomatas thus effective ozone uptake." To stress this temperature dependence, we added some sentences in the methods section from which implies that the temperature increase leads to increase in dry-deposition. The same was done for BVOC emissions, as these are also strongly dependent on near-surface temperatures.

•If the piece about increasing in ozone due to suppressed dry dep is not shown in this manuscript, discussion is not merited in the intro. Too speculative.

Authors response: Figure 7 clearly shows that ozone dry-dep is decreased due to urbanization related land-cover change so we can say the increase of ozone is probably due to this suppression.

Table 1.

•I think it would be clearer to say 'crop' rather than 'Nourban'

Authors response: We prefer to use "Nourban" to be consistent with our previous studies and with the fact that it represent the land-surface without urbanization, i.e. rural land-surface. However, we added explicitly in the footnote that Nourban means crops.

Table 2.

•For parallel structure, do Landuse (deposition) vs. Landuse (BVOC) as the column names.

Authors response: corrected. We also changed the last column name to Driving meteorology (BVOC)

•It's confusing to have 'nourban vs urban' for meteorology, and then 'rural vs. urban' for land use. Do the authors want to have different descriptors here? If so, please spell out why in the table footnotes and/or text.

Authors response: Urban/Nourban for meteorology more relates to the effects, i.e. the differences in the meteorological parameters between the Urban and Nourban landcover. In Tab 2 therefor we prefer to use Urban/Nourban to columns were we refer to the driving meteorology (to be consistent with Tab 1.). On the other hand, in the "Landuse" columns, we would like to stress that the landuse is "rural". So we prefer to use different descriptors.

**Figure 1.**

•Can this map include the model grid? (Also, are the two model grids the same?)

Authors response: the map exactly corresponds to model grid (model domain) which is same for WRF/RegCM/CAMx. Adding the gridlines would make the figures very messy.

•Make city symbols larger.

Authors response: Corrected.

•What does the [m] in the title mean?

Authors response: Means "meters". Corrected.

•Why show terrain rather than land use type?

Authors response: we agree that showing the urban landuse would be logical, however the main message from such a figure would be the location of the main urban centres, which is indicated in the actual figure too. Moreover, the urban landuse figure was already presented in Karlicky et al. 2020 (Fig 1 and 2) which uses the same domain and landuse data. So we rather decided to show a different information: the orography which is an important parameter modulated the regional climate. We explicitly refer to this paper in the revised manuscript if the reader is interested in the landuse information.

**Figure 2**

•Are the observations from the same year as the model? Please specify

Authors response: of course, observation are from the same date/time as the model data. Clarified in the manuscript.

•Can the authors put error bars on the observations to represent variability across stations examined?

Authors response: as the WRF driven CAMx "lines" were added to the figures (see the response below), we think that the errorbars would make the (sub)figure(s) harder to read. Also, the data are from small amount of stations per city (usually 1 to 10 station per city) to make the errorbars representative. We therefore prefer the presentation without errorbars.

•Why is there only one model line on here? Shouldn't there be two (the CAMx simulations with different meteorology)?

Authors response: in the revised manuscript the WRF driven CAMx simulations was added to the validation (green).

•I get that some authors like to convert the model values to whatever units the observations are in for model evaluation, but the inconsistency between the units of this plot and the ppbv used elsewhere makes contextualizing the model's limitations challenging.

Authors response: it is true that the units for the results (ppbv) are different from the units in the validation (ug/m3) however we preferred to stay consistent with previous validation studies (ours and others too) where the model values were converted to the units of the measurements. Morever, in this conversion we had to use the modeled temperature and pressure values (needed for the ppbv  $\leftrightarrow$  ug/m3 conversion). However, we do not have the temperature/pressure data from the stations chosen, so we cannot correctly convert the measured data into ppbv, only vice-versa.

**Figure 3**

•The error bars show the maximum and minimum across grid cells, or across days?

Authors response: they show the 25% and 75% percentiles as well as the minimum/maximum values across all the chosen cities. Clarified in the revised manuscript.

**Figures 4,5,6 and 8**

•Can the Prague panel be the first panel, and then the larger domain ones be the second and third panels? This would make it easier to compare the two larger domain panels...

Authors response: We agree with the referee and reordered the columns so now the first one shows the detailed result for Prague. We adapted the captions and the text accordingly.

**Figure 10**

•Are these changes, or absolute? I think the former because there are some negatives. These changes look huge, but I don't know what the baseline is. Maybe just show the individual simulations (rather than the changes)? This goes back to my saying that you can't really consider the differences in the simulations as additive.

Authors response: these are the individual contributors to the total urbanization impact (defined by Eq.5-8), so we present differences of simulations (that's why we have negative values, indeed). There is no single baseline but the contributions are built on each other as mentioned earlier and defined by these equations. So by this definition these contributors are additive. The reason for not presenting the absolute values is that the difference between the simulations (defining these contributors) is often very small. This is the reason, why we have chosen a separate y-axis for the weaker contributors (DLU and DBVOC; the right axis).

**Figure 11**

•It might be better to show both DV for urban vs. crop, because it's hard to contextualize the impact without knowing the absolute values (& many readers may be unfamiliar with the magnitude of DV.

Authors response: in the revised manuscript, we added the absolute DVs corresponding to rural landuse (crops) as solid lines (left y-axis) and kept the differences arising from urbanization (i.e. the impact of urban landuse; dashed lines; right y-axis). Now the reader easily seas the difference and compare it to the absolute values.

•Please convert to mm/s which is more widely used for DV than mm/h & agrees with the units used on the other plot on which you have DV.

Authors response: The units for DV are mm/s everywhere. There was a mistake in the figure caption. Corrected.